# Budgerigars have complex sleep structure similar to that of mammals

**Sofija V. Canavan**[1,2]*, **Daniel Margoliash**[1,3]

**1** Committee on Computational Neuroscience, University of Chicago, Chicago, Illinois, United States of America, **2** Medical Scientist Training Program, University of Chicago, Chicago, Illinois, United States of America, **3** Department of Organismal Biology and Anatomy, University of Chicago, Chicago, Illinois, United States of America

* svcanavan@uchicago.edu

**Data Availability Statement:** All relevant data are within the paper and its Supporting Information files.

**Funding:** This work was supported by the National Institutes of Health grants T32 GM007281 and F30 MH113298 (to S.V.C.; National Institute of General

## Abstract

Birds and mammals share specialized forms of sleep including slow wave sleep (SWS) and rapid eye movement sleep (REM), raising the question of why and how specialized sleep evolved. Extensive prior studies concluded that avian sleep lacked many features characteristic of mammalian sleep, and therefore that specialized sleep must have evolved independently in birds and mammals. This has been challenged by evidence of more complex sleep in multiple songbird species. To extend this analysis beyond songbirds, we examined a species of parrot, the sister taxon to songbirds. We implanted adult budgerigars (*Melopsittacus undulatus)* with electroencephalogram (EEG) and electrooculogram (EOG) electrodes to evaluate sleep architecture, and video monitored birds during sleep. Sleep was scored with manual and automated techniques, including automated detection of slow waves and eye movements. This can help define a new standard for how to score sleep in birds. Budgerigars exhibited consolidated sleep, a pattern also observed in songbirds, and many mammalian species, including humans. We found that REM constituted 26.5% of total sleep, comparable to humans and an order of magnitude greater than previously reported. Although we observed no spindles, we found a clear state of intermediate sleep (IS) similar to non-REM (NREM) stage 2. Across the night, SWS decreased and REM increased, as observed in mammals and songbirds. Slow wave activity (SWA) fluctuated with a 29-min ultradian rhythm, indicating a tendency to move systematically through sleep states as observed in other species with consolidated sleep. These results are at variance with numerous older sleep studies, including for budgerigars. Here, we demonstrated that lighting conditions used in the prior budgerigar study—and commonly used in older bird studies—dramatically disrupted budgerigar sleep structure, explaining the prior results. Thus, it is likely that more complex sleep has been overlooked in a broad range of bird species. The similarities in sleep architecture observed in mammals, songbirds, and now budgerigars, alongside recent work in reptiles and basal birds, provide support for the hypothesis that a common amniote ancestor possessed the precursors that gave rise to REM and SWS at one or more loci in the parallel evolution of sleep in higher vertebrates. We discuss this hypothesis in terms of the common plan of forebrain organization shared by reptiles, birds, and mammals.

Medical Sciences, https://www.nigms.nih.gov;
National Institute of Mental Health, https://www.
nimh.nih.gov/index.shtml) and R01 DC012859 (to
D.M.; National Institute of Deafness and Other
Communication Disorders, https://www.nidcd.nih.
gov). The funders had no role in study design, data
collection and analysis, decision to publish, or
preparation of the manuscript.

**Competing interests:** The authors have declared
that no competing interests exist.

**Abbreviations:** EEG, electroencephalogram; EMG,
electromyography; EOG, electrooculogram; FP,
false positive; IS, intermediate sleep; LD, light/dark;
LL, constant light; nPeaks, number of peaks per
epoch;  N2, non-REM stage 2 sleep; NREM, non-
REM; REM, rapid eye movement sleep; PSG,
polysomnography; SWA, slow wave activity; SWS,
slow wave sleep; TN, true negative; TP, true
positive; TST, total sleep time.

## Introduction

Although sleep is broadly expressed [1,2], a subset of species have developed specialized forms of sleep. Rapid eye movement sleep (REM) and slow wave sleep (SWS), with their associated changes in cortical activation, are found only in mammals and birds [3–8]. Many hypotheses of sleep in mammals ascribe specific functions to REM, SWS, and features thereof [9–14]. Understanding how and why complex sleep architecture evolved in birds may inform how we understand sleep evolution as a whole [15–17] and will constrain hypotheses on the role of sleep that have focused on mammals and neocortex.

While the structure of sleep in mammals is extensively studied—especially in humans [18] —and conforms to a broadly accepted general pattern, the description of sleep in birds is undergoing rapid revision. A wealth of older studies in birds found only 2%–7% REM [reviewed in 4,6,19], inconsistent ultradian regulation [20–24], and no SWS rebound after sleep deprivation [25]. This promoted the hypothesis that sleep evolved independently in birds and mammals and shares few functional similarities.

More recently, complex mammalian-like sleep with abundant REM has been observed in multiple songbird species [26–28], and may even extend to certain species of basal birds [29,30]. Ostriches, ratites of the avian clade Paleognathae, exhibit a hybrid REM state [29] that combines some attributes of SWS and strongly resembles hybrid REM in monotremes [31,32], the most basal group of mammals. In contrast, tinamous, another paleognath species, have typical REM [30] like that of neognaths. Therefore, it remains unresolved whether hybrid REM evolved independently in ostriches and monotremes, and whether "normal" REM dates back to the common ancestor of extant birds, or perhaps extant Neognathae (which includes the vast majority of living birds).

This gives rise to several questions: how to reconcile the old and new findings, how broadly distributed in birds the mammalian-like sleep structure is, and how this impacts theories of sleep mechanisms and evolution. The role of sleep in birdsong learning [33–36], in adult birds acquiring perceptual memories [37], and in maintaining memories through cycles of reconsolidation [38] all share striking similarities with observations in mammals [39]. This further motivates interest in understanding how similarities in sleep arose.

Here, we investigated sleep architecture in a parrot species, budgerigars (*Melopsittacus undulatus*). Parrots (Psittaciformes) are the sister taxon to songbirds [40,41] and, like songbirds, are one of 3 orders of birds that possess vocal learning abilities [42,43]. We found that budgerigars have abundant REM, a distinct non-REM stage 2 (N2)-like state of intermediate sleep (IS), and circadian and ultradian rhythms in sleep structure that mirror those found in songbirds and mammals. We showed that the budgerigar sleep/wake cycle is significantly altered by constant light, which was used to facilitate observations of sleeping birds in several early sleep studies prior to the advent of widespread availability of infrared video technology, including the only previous study of budgerigar sleep [23]. This led to considerable disruption of sleep in budgerigars, and likely does so in other species. Finally, our results help define an emerging framework for scoring sleep in birds, taking into account the newer sets of observations. A standard for defining sleep architecture in birds may provide benefits comparable to what has been enjoyed in mammalian studies [18,44].

## Results

### Characteristics of budgerigar sleep behavior

We first sought to describe the sleeping behavior of the budgerigars. All birds exhibited a diurnal pattern of activity. Lights off was invariably followed by 10–20 min of vigorous activity as

birds moved around the enclosure, sometimes circling the perimeter several times before settling into their preferred spot. At night, all 5 birds frequently climbed up the sides and sometimes onto the ceilings of their enclosures to sleep. One bird (Bird 4) was observed sleeping for several hours while hanging upside down from the ceiling (Fig 1A). During the day, climbing was almost never observed, except early in the morning when birds might remain in their nighttime sleeping position for up to an hour. Although birds napped often during the day, they usually remained perched near the floor of the enclosure.

At night, birds engaged in long bouts of deep sleep (deep rhythmic breathing, muscle twitches, and head drooping) punctuated by brief awakenings of a few seconds. Most birds also exhibited a few longer nighttime awakenings during which they either climbed to a new sleeping position or moved to the food dish and ate. In contrast, behaviorally apparent deep sleep with noticeable deep rhythmic breathing or other features occurred very infrequently during the day. Most daytime naps occurred as brief sleep episodes amid long periods of drowsiness. Overall birds slept far more at night (82.9% ± 6.9% of time) than during the day (17.4% ± 11.9% of time; $t(4) = 9.40$, $p = 0.0007$) (Tables 1 and 2).

## Electrophysiological characteristics of budgerigar sleep

To further characterize budgerigar sleep, we examined the polysomnography (PSG) data. Active wake contained large bouts of movement artifacts as expected. Quiet wakefulness revealed a low-amplitude and high-frequency EEG with frequent eye movements in the EOG (Fig 1B, Wake). Eye movements sometimes caused large artifacts in the EEG which could resemble delta waves (Fig 1B, Wake). During drowsiness, it was common for slower and higher-amplitude elements to appear in the EEG, while the EOG reflected frequent slow eye movements and blinking (Fig 1B, Drowsy).

Unihemispheric sleep was observed wherein the hemisphere contralateral to the closed eye exhibited low-frequency EEG activity resembling either IS or SWS, while the other hemisphere continued to show wake-like activity (Fig 1C). Notably, birds exhibited more unihemispheric sleep during the day (Table 1, Fig 2E and 2F). This study was not specifically designed to maximize observation of unihemispheric sleep, and due to the unanticipated mobility of budgies during sleep, it was difficult to visualize both eyes simultaneously using 1 camera. We scored unihemispheric sleep only during periods in which both eyes were visible; therefore, the total amount of unihemispheric sleep was most likely underestimated. The unihemispheric sleep we observed tended to occur in very short bouts (4.8 ± 0.57 s; maximum duration in any given individual ranged from 18 to 42 s) and most often resembled drowsiness in posture, breathing, and the appearance of the open eye.

Strikingly, budgerigars had large amounts of REM, characterized by fast-frequency low-amplitude EEG activity and large rapid eye movements appearing in the EOG (Fig 1D). We observed both phasic REM with frequent eye movements (Fig 1D) and tonic REM with few eye movements (Fig 1G). Eye movements that were visible on camera were rare but did occur (S1 Video); for example, in Bird 1, in which eye movements were most reliably observed, 42/51 visible eye movements during sleep occurred during REM.

We also identified large amounts of IS, a non-REM (NREM) non-SWS state (Fig 1E). Consistent with prior work, we observed SWS episodes with long continuous trains of slow waves (Fig 1F). In contrast with SWS, IS typically contained a mix of low-amplitude delta, high-amplitude K-complex-like waves (Fig 1H), and higher-frequency elements such as theta (Fig 1E). Birds often alternated quickly between IS and SWS as slow wave content fluctuated (Fig 1I). We did not identify sleep spindles in budgerigars (see Discussion).

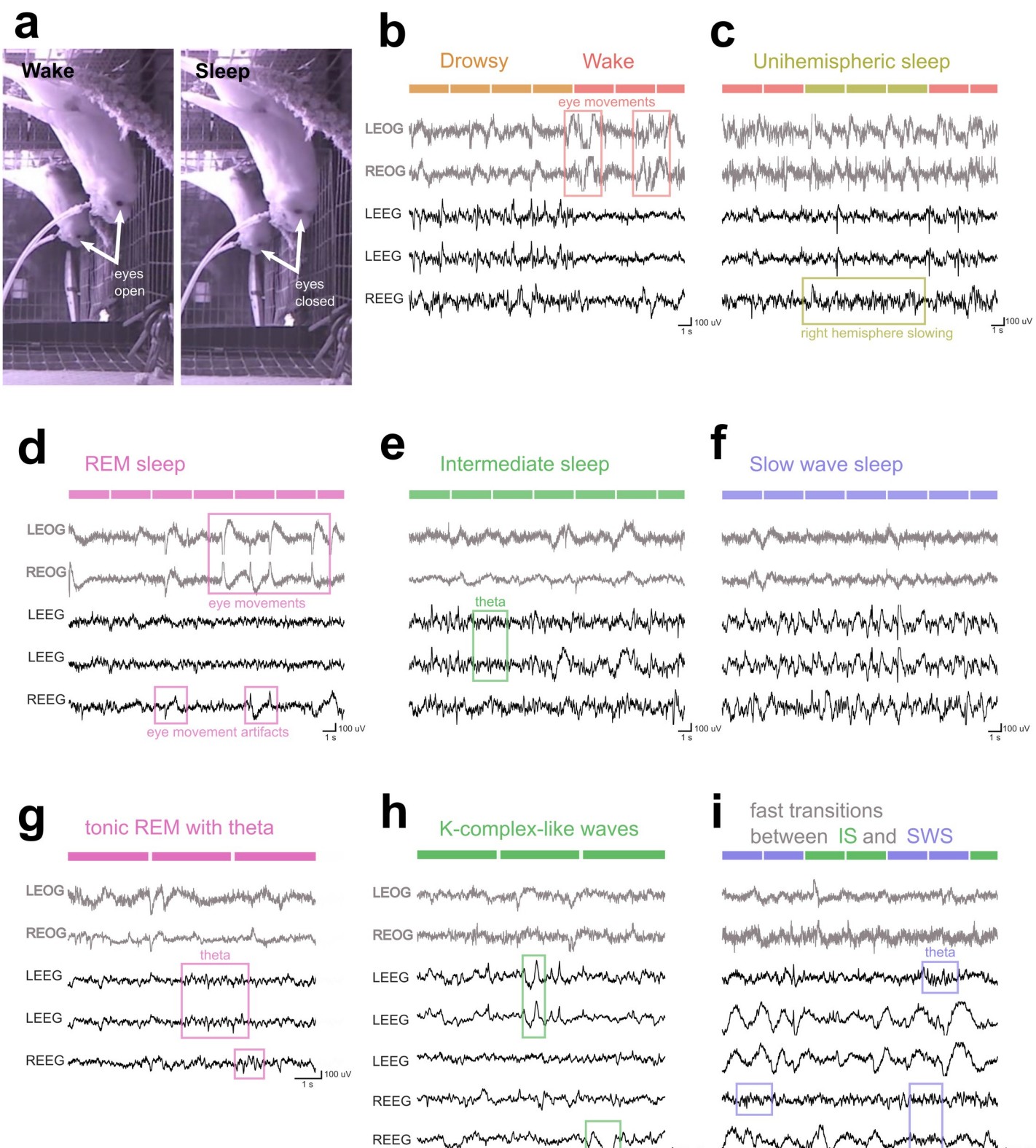

**Fig 1. Examples of video and PSG data.** (a) Infrared image of Bird 4 falling asleep at night while hanging from the ceiling of the enclosure. The bird is facing the cage wall on the right of the picture. A mirror on the back wall reveals the eye contralateral to the camera. During the brief awakening (left), both eyes were open; contrast with closed eyes (right) during sleep (2 images separated by approximately 6 min). (b–h) Example PSG recordings. The upper 2 EOG traces are gray; the lower EEG

traces are black, 3 artifact-free traces for Bird 1 (panels b–g) and 5 artifact-free traces for Bird 4 (panels h–i). Within each hemisphere, EEG channels are arranged from most anterior at the top to most posterior at the bottom. Note smaller time base in panels g–h to highlight details of the EEG recording. (b) A transition from drowsiness to full wakefulness. During drowsiness, note the occasional (every 1–2 s) slow, medium-amplitude EEG oscillations occurring along with higher-frequency elements. As the bird rapidly transitions back to alert wakefulness, the EEG rapidly transitions to low amplitude and high frequency. Frequent eye movements appear on the EOG throughout both states, but especially during wake. (c) Unihemispheric sleep. Medium-amplitude delta activity (0.5–4 Hz) appears in the right hemisphere, while the left hemisphere maintains a flat, wake-like EEG. The left eye was closed, and the right eye was open. (d) REM. Note wake-like EEG with bursts of fast eye movements. Two eye movement artifacts are identified in the right EEG channel. (e) IS. Low-frequency, medium-amplitude oscillations appear in the EEGs of both hemispheres. These are larger than either REM or wake EEGs (cf. 1b, 1d), but only infrequently reached an amplitude sufficient to be classified as slow waves. An example of theta (4–8 Hz) is highlighted. (f) SWS. The EEG contains numerous slow waves (0.5–4 Hz, >4 times wake amplitude). (g) A long period (9 s) of tonic REM with few eye movements and a particularly clear example of theta. (h) An example of K-complex–like waves observed during IS. (i) Example using the amount of SWA to distinguish between SWS and IS. The first IS epoch contained one slow wave in the last channel, preventing this epoch from being scored as REM. This example also includes instances of theta during SWS. EEG, electroencephalogram; EOG, electrooculogram; IS, intermediate sleep; PSG, polysomnography; REM, rapid eye movement sleep; SWA, slow wave activity; SWS, slow wave sleep.

**Table 1. Sleep stage proportions.**

| Nighttime | | | | |
|---|---|---|---|---|
| **Stage** | **% of Recording Time** | | **% of TST** | |
| | **Mean** | **SD** | **Mean** | **SD** |
| **Wake** | 11.98 | 6.34 | - | |
| **Drowsy** | 5.09 | 1.62 | - | |
| **Unihemispheric sleep** | 0.023 | 0.019 | 0.028 | 0.023 |
| **IS** | 42.24 | 11.20 | 50.44 | 10.22 |
| **SWS** | 15.70 | 5.10 | 18.86 | 5.65 |
| **REM** | 24.96 | 7.94 | 30.68 | 11.52 |
| **TST** | 82.93 | 6.88 | - | |
| **Daytime** | | | | |
| **Stage** | **% of Recording Time** | | **% of TST** | |
| | **Mean** | **SD** | **Mean** | **SD** |
| **Wake** | 49.82 | 15.29 | - | |
| **Drowsy** | 32.80 | 14.76 | - | |
| **Unihemispheric sleep** | 3.27 | 2.35 | 29.33 | 21.88 |
| **IS** | 9.32 | 7.29 | 51.66 | 6.84 |
| **SWS** | 3.12 | 3.81 | 13.00 | 10.40 |
| **REM** | 1.67 | 2.44 | 6.01 | 6.96 |
| **TST** | 17.37 | 11.89 | - | |
| **24 Hours** | | | | |
| **Stage** | **% of Recording Time** | | **% of TST** | |
| | **Mean** | **SD** | **Mean** | **SD** |
| **Wake** | 32.55 | 10.44 | - | |
| **Drowsy** | 20.09 | 7.56 | - | |
| **Unihemispheric sleep** | 1.77 | 1.21 | 3.83 | 2.43 |
| **IS** | 24.42 | 5.29 | 51.48 | 7.46 |
| **SWS** | 8.90 | 3.95 | 18.18 | 5.82 |
| **REM** | 12.26 | 3.55 | 26.51 | 8.97 |
| **TST** | 47.35 | 6.68 | - | |

The proportions of the 6 behavioral stages (wake, drowsy, and 4 sleep states) are presented as a function of percentage of recording time and of TST over the nighttime, daytime, and over the total 24-h period.

**Abbreviations:** IS, intermediate sleep; REM, rapid eye movement sleep; SWS, slow wave sleep; TST, total sleep time

**Table 2. Nighttime sleep of individuals.**

| Individual Birds | Bird 1 | Bird 2 | Bird 3 | Bird 4 | Bird 5 |
|---|---|---|---|---|---|
| IS (% TST) | 33.41 | 51.29 | 59.16 | 50.79 | 57.54 |
| SWS (% TST) | 19.06 | 12.23 | 22.33 | 26.12 | 14.53 |
| REM (% TST) | 47.52 | 36.42 | 18.47 | 23.09 | 27.90 |
| TST (% recording time) | 76.99 | 79.24 | 94.25 | 79.78 | 84.41 |

For each individual bird, the 3 main sleep states are shown as a percentage of TST for the nighttime. TST is shown as a percentage of the total recording time over the night.

**Abbreviations:** IS, intermediate sleep; REM, rapid eye movement sleep; SWS, slow wave sleep; TST, total sleep time

We occasionally saw instances of head drooping as the budgerigars slept. Upon reviewing the behavioral notes—which were made prior to PSG scoring—we found that the majority of these instances occurred during REM, although the great majority of REM was not accompanied by head drooping. Only 8.6 ± 3.1 overt instances of head drooping were recorded per bird (S1 Data); of these, roughly half (48.3% ± 17.2%) were later found to have occurred during drowsiness. Of those during sleep, 52.5% ± 18.4% instances of drooping occurred during REM and 39.6% ± 19.1% during IS. The head drooping appeared similar to the description of slow, controlled dropping of the head reported during REM muscle hypotonia in geese [45], rooks [22], chickens [46], white-crowned sparrows [47], and tinamous [30].

A broad variety of species express twitching during sleep [48], particularly REM [49]. Twitching was a very common occurrence during budgerigar sleep: across birds, 220.4 ± 113.0 episodes of twitching during sleep were recorded (range: 44 to 643 twitching episodes/bird). Of these, 50.1% ± 5.6% began during REM epochs, while 35.5% ± 3.9% began during IS epochs (S1 Data), despite most birds spending more time in IS than REM (Table 1, Fig 2E–2G). Thus, both head drooping and twitching were primarily associated with REM but also seen frequently during IS. Three of the birds also exhibited repetitive beak movements during sleep, but these were reliably observed only in Bird 1. Of the 152 beak movements observed in that bird, 101 occurred during IS and 37 during REM (S1 Data). This was despite approximately equal amounts of 24-h REM and IS in this bird (38.6% and 38.7%, Fig 2G; also see Table 2 for nighttime sleep). Thus, beak movements, unlike muscle twitches or head drooping, may be more prevalent during IS.

In mice and rats, hippocampal theta (4–8 Hz) is prominent in the EEG during REM and wake. In humans, theta waves are less apparent on the surface EEG but can appear as bursts of sawtooth waves during REM. While we did observe occasional REM theta in the budgerigars (Fig 1G), theta bursts could also appear during IS (Fig 1E) or during SWS (Fig 1I). Thus, we did not find theta to be a useful indicator of sleep stage in budgerigars. This is consistent with findings from depth recordings from chicken hippocampus [50].

## Sleep architecture across day and night

We next considered how budgerigar sleep architecture was modulated over 24 h. Individual hypnograms revealed considerable fast fluctuations between vigilance states (Figs 2A and S1), but several consistent patterns emerged (Fig 2B). During the day, birds tended to remain either awake or drowsy, with frequent but short bouts of IS and unihemispheric sleep (Fig 2A and 2B). Napping was most common in the early afternoon, resulting in a small dip in wakefulness around 7 h after lights on (Figs 2A, 2B and S1). After lights off, birds fell asleep during the first hour and slept largely in a consolidated block until shortly before lights on (Fig 2B).

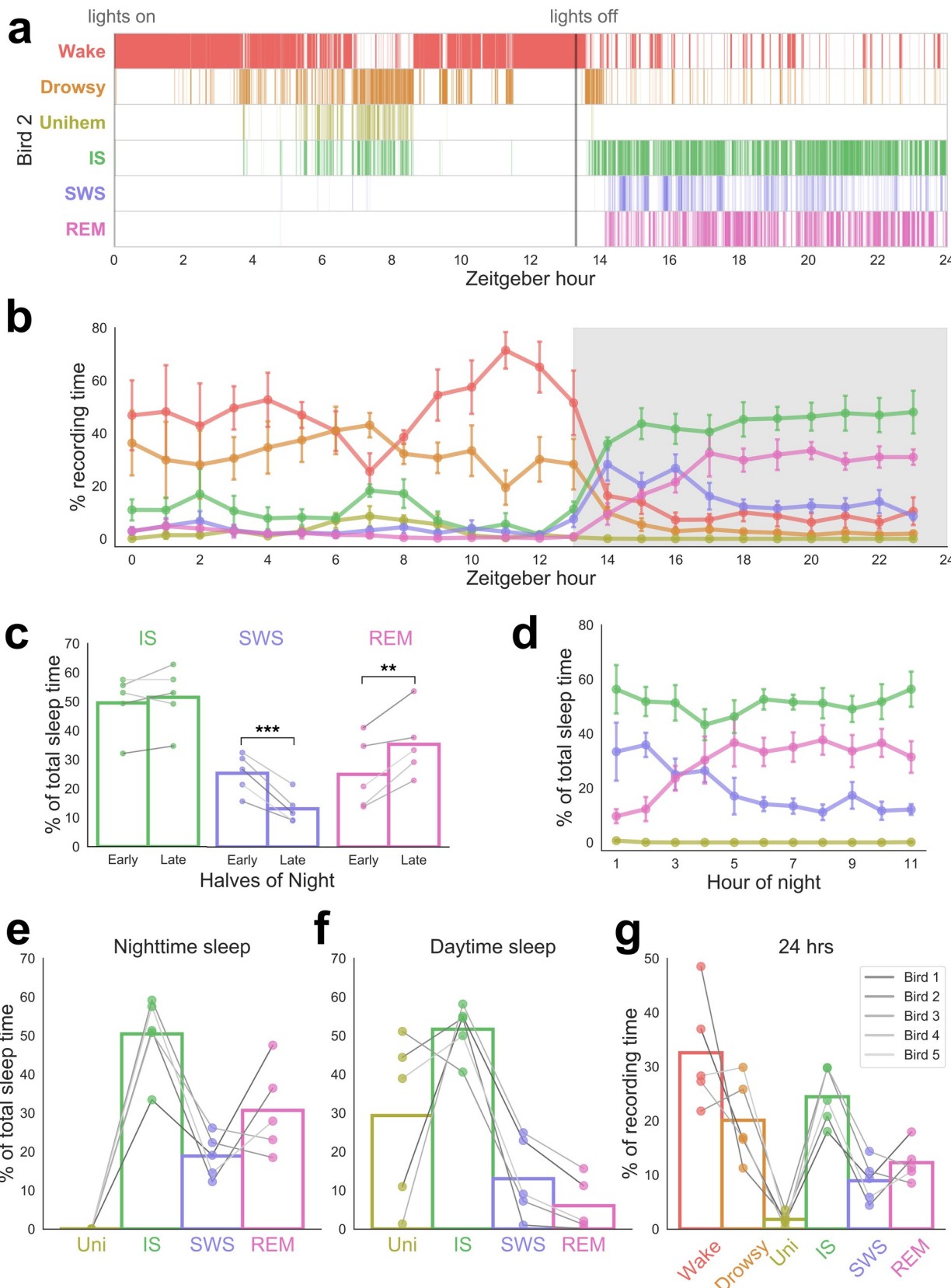

**Fig 2. Sleep architecture and rhythms over 24 h.** (a) An example hypnogram showing sleep scored across 24 h in a single bird (Bird 2). Each tick represents one 3-s epoch with rows corresponding to vigilance states. Time of lights on and lights off are indicated by grey vertical lines. (b) The mean ± SEM of vigilance states across the 24-h recording period in all birds. Lights off is shaded in grey. (c) Comparison across the 2 halves of the night. Sleep states are shown as a percentage of TST. Paired $t$ tests: $^{**}p < 0.01$, $^{***}p < 0.005$. (d) Hour-by-hour patterns of sleep states as a percentage of TST. (e–f) Total amounts of each sleep state as a percentage of TST during the night and day. (g) Total amounts of each vigilance state across 24 h. In e–g, each point represents one bird, and bars show the mean. Lines in c and e–g are shaded according to bird identity, ranging from Bird 1 in the darkest shade to Bird 5 in the lightest. Data are provided in S1 Data. Raw scores are provided in S2 Data. IS, intermediate sleep; REM, rapid eye movement sleep; SWS, slow wave sleep; TST, total sleep time; Uni, unihemispheric sleep; Unihem, unihemispheric sleep.

To look for patterns of changes across the night, we divided the night in half and examined each stage of sleep as a percentage of total sleep time (TST) (Fig 2C). From the first to the second half of the night, REM significantly increased ($t(4) = -5.00$, $p = 0.007$), SWS significantly decreased ($t(4) = 7.18$, $p = 0.002$), and IS did not change significantly ($t(4) = -1.05$, $p = 0.35$) (Fig 2C).

When we considered sleep stages across the night on an hour-by-hour basis (Fig 2D), the same pattern was observed wherein REM increased (regression with hour of night: slope = 2.27% TST/h, $r^2 = 0.589$, $p = 0.006$), SWS decreased (slope = −2.37% TST/h, $r^2 = 0.770$, $p = 0.0004$), and IS remained the same (slope = 0.13% TST/h, $r^2 = 0.013$, $p = 0.74$). Results were similar when sleep stages were regressed not with time but with hour of total sleep (REM: slope = 2.71%, $r^2 = 0.673$, $p = 0.007$; SWS: slope = −2.67%, $r^2 = 0.672$, $p = 0.007$; IS: slope = −0.034, $r^2 = 0.00053$, $p = 0.95$). These patterns appeared in all individual birds (Fig 2C), including those for which sleep was scored blind to hour of night.

The total amounts of sleep stages expressed by a given species are thought to provide important clues as to how sleep co-evolved with other species characteristics [5,51,52]. Comparative studies of sleep in animal species typically consider sleep across a 24-h recording period. In contrast, human sleep statistics are typically reported for nighttime sleep only. To facilitate comparison, we report both measures here (Table 1, Fig 2E–2G). Nighttime sleep was composed of large amounts of REM (30.7% ± 11.5% of TST) and 18.9% ± 5.6% SWS, with IS occupying about half of TST (50.4% ± 10.2%) (Fig 2E; see Table 1 for values for all sleep stages; see Table 2 for nighttime values of all individual birds).

Sleep architecture during the day (Fig 2F) was markedly different from nighttime sleep. Most of this limited daytime sleep consisted of either IS (51.6% ± 6.8% of daytime TST) or unihemispheric sleep (29.3% ± 21.9%). Unihemispheric sleep was significantly higher than at night ($t(4) = 3.00$, $p = 0.04$). REM made up the smallest share of daytime TST at only 6.0% ± 7.0% and was significantly lower than at night ($t(4) = -4.42$, $p = 0.01$). In total across a 24-h period, animals spent 11.4 ± 1.6 h asleep.

## Fine-scale structure of sleep states

Many previous studies have described avian REM as particularly unstable, occurring in very short episodes. We therefore examined the continuity of each sleep state. Across the night, REM episodes became longer in duration (Fig 3A; regression with hour of night: slope = 0.80 s/hour, $r^2 = 0.765$, $p = 0.0004$), as did IS episodes (slope = 0.36 s/hour, $r^2 = 0.681$, $p = 0.002$), whereas SWS episodes became slightly shorter (slope = −0.24 s/hour, $r^2 = 0.708$, $p = 0.001$). All 3 sleep stages were relatively brief in duration: REM episodes were the longest and most variable (11.3 ± 5.1 s), IS episodes were shorter (9.2 ± 1.3 s), and SWS episodes were the shortest (5.1 ± 0.8 s). A one-way ANOVA indicated significant differences in the duration of the different sleep stages ($F(4) = 4.21$, $p = 0.04$), which resulted from SWS episodes being significantly shorter than IS episodes ($t(4) = -4.7$, $p = 0.01$) and trending shorter than REM ($t(4) = -2.59$, $p = 0.06$). The duration of IS and REM episodes did not differ ($t(4) = -0.689$, $p = 0.53$). These

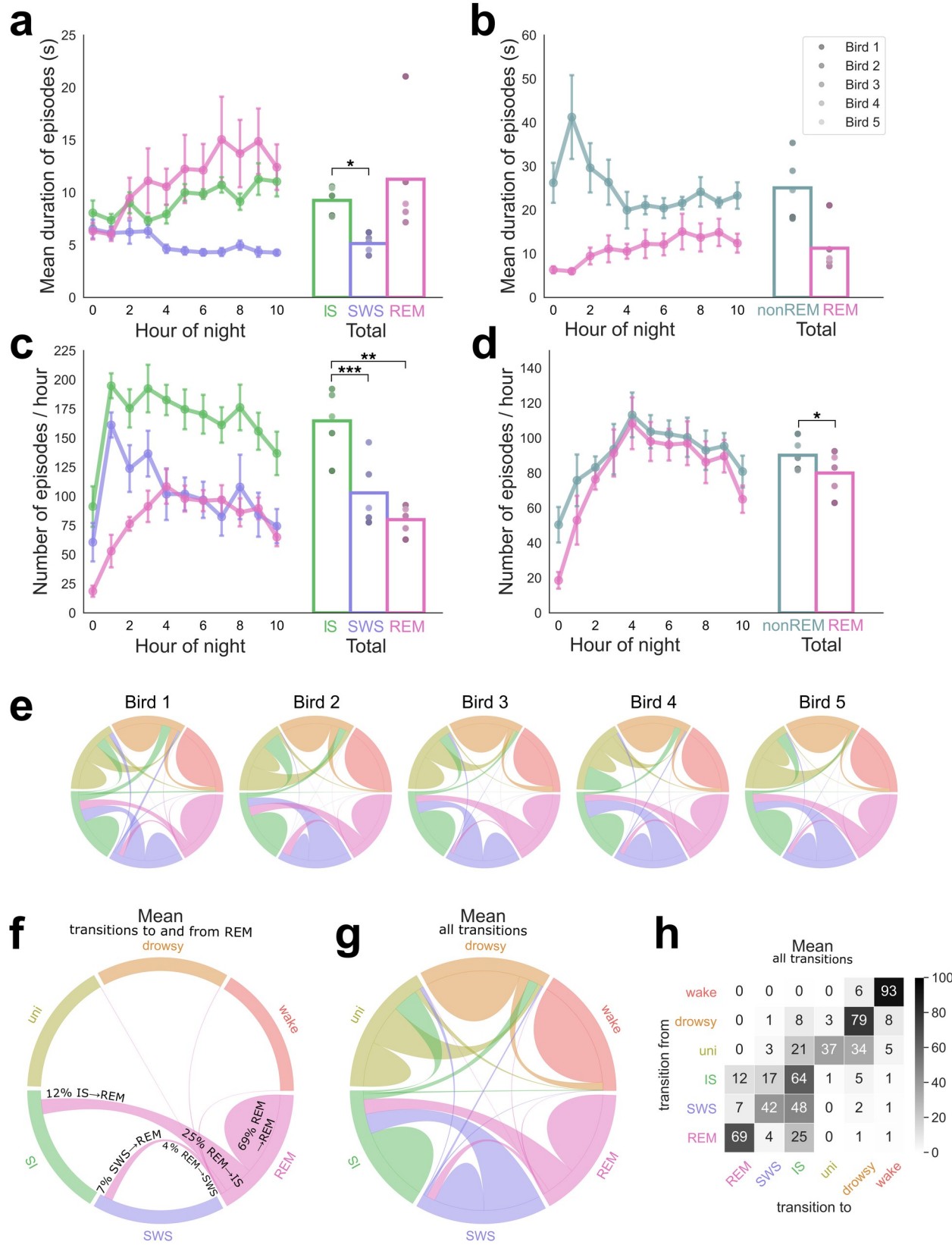

**Fig 3. Sleep state durations, frequencies, and transitions.** (a) Durations of each sleep state. Left: durations per hour of night. Right: mean durations across the entire night. (b) Durations of sleep states with IS and SWS combined into a single NREM state. Note that IS and SWS frequently combine to make longer episodes, so by this classification scheme REM episodes appear markedly shorter. (c) The frequency of episodes of each sleep state, both per hour of night (left) and the mean over the whole night (right). (d) The frequency of episodes of each sleep state, with IS and SWS combined into a single NREM state, making fewer, longer episodes. Dots in a–d are shaded according to bird identity, ranging from Bird 1 in the darkest shade to Bird 5 in the lightest. Paired t tests: *$p < 0.05$; **$p < 0.01$; ***$p < 0.005$. (e) Chord diagrams showing transitions between sleep/wake states for each individual bird. Note the similarity in the pattern across all 5 individuals. (f) Chord diagram with detail of the mean across all birds displaying only transitions to and from REM episodes. Each of the 6 sleep/wake states is represented as an equal "slice" on the outer circle. Each line or "chord" between 2 states represents the transitions between these states, with the line thickness at each of the 2 bases representing the frequency of the transition from that base to its target. A "hump" confined to a single state corresponds to self-transitions, e.g., REM to REM. Note most transitions in and out of REM tend to be to/from IS rather than SWS. (g) Chord diagram of the mean across all birds, displaying all transitions. The most common transitions are between the 3 bihemispheric sleep states (IS, SWS, and REM) and between the other 3 states (wake, drowsy, and unihemispheric sleep). IS serves as the most common link between these 2 categories. (h) Transition matrix, displaying the same data as f–g. Numbers (rounded to integers) and shading correspond to percentage of transitions out of a given state. Data are provided in S1 Data. IS, intermediate sleep; NREM, non-REM; REM, rapid eye movement sleep; SWS, slow wave sleep; Uni, unihemispheric sleep.

observations underscore that all the sleep stages in budgerigars tended to occur in short episodes, and that REM was expressed reliably during sleep.

Because sleep studies in animal models often do not divide NREM into SWS and IS, we also examined NREM as a whole. The duration of NREM episodes was highest in the second hour of the night and then fell sharply, reaching a plateau after 4 h (Fig 3B). Episode durations were low in the first hour of the night, which contained overall low TST (35.9%) as birds climbed the walls and explored the cage for considerable periods of time, reducing all the measures correlated with sleep. Birds were well into their consolidated sleep patterns by the second hour of the night. Thus, only when the first hour was omitted did NREM episode duration display a significant linear decrease across the night (slope = −1.36 s/hour, $r^2 = 0.405$, $p = 0.048$). On average, NREM episodes ($25.1 \pm 6.6$ s) were longer than REM episodes ($11.3 \pm 5.1$ s) (Fig 3B) although this difference did not quite achieve significance (paired t test, $t(4) = 2.57$, $p = 0.06$). In this regard, we note that one animal (Bird 1) exhibited approximately twice the average REM episode duration of the other 4 birds (Fig 3A and 3B).

The frequencies of episodes also differed between sleep stages (one-way ANOVA, $F(4) = 16.04$, $p = 0.0004$), with IS episodes occurring more frequently than either SWS ($t(4) = 7.71$, $p = 0.002$) or REM ($t(4) = 7.31$, $p = 0.002$). SWS and REM episodes did not significantly differ in frequency. The pattern in the frequency of episodes of SWS and REM roughly mirrored the durations, with SWS decreasing and REM increasing (Fig 3C). Conversely, the frequency of IS decreased, the opposite of the trend in IS duration; this explains the consistent amount of IS throughout the night. These trends in IS and SWS were not significant, but as per above this was due to the low TST in the first hour of night: when this hour was omitted, there was a highly significant linear decrease in both IS (slope = -4.90 s/hour of night, $r^2 = 0.732$, $p = 0.002$) and SWS (slope = −7.66 s/hour of night, $r^2 = 0.753$, $p = 0.001$).

When IS and SWS were combined into NREM, the frequency of episodes closely followed those of REM. This is to be expected in consolidated sleepers; in other words, birds largely alternated between REM and NREM with few transitions into wake, resulting in extremely similar patterns in the number of REM and NREM episodes. There were significantly more NREM episodes ($t(4) = 4.15$, $p = 0.014$) likely arising from NREM-only sleep episodes especially toward the beginning of the night.

We also quantified transitions between sleep/wake states. The pattern of transitions was remarkably similar across the 5 individual birds (Fig 3E). Transitions in and out of REM were of particular interest (Fig 3F). IS was the most common state either preceding or following REM. Less frequently, REM was preceded by SWS. A small number of REM episodes were followed by SWS. Transitions between REM and any other states were exceedingly rare. In general, once asleep birds tended to stay asleep, with the most common transitions occurring

either between IS, SWS, and REM or between wake, drowsy, and unihemispheric sleep (Fig 3G and 3H). The most prominent bridges spanning awake states and asleep states were IS-drowsiness transitions; birds tended not to wake up directly from either REM or SWS.

## Spectral characteristics and slow wave activity

We computed power spectra to characterize frequency content (Fig 4A–4C) and Rüger's areas to assess significance of differences between sleep states (Fig 4C). This confirmed the manual scoring, which relied largely on delta content (Fig 4A and 4C), and revealed additional differences in higher frequency bands less visible to the manual scorer (Fig 4B and 4C). Compared to other vigilance states, SWS exhibited the highest amount of low-frequency power with a broad peak encompassing the delta band (Fig 4C). SWS significantly exceeded REM in the frequency range 1.67–4 Hz, which spans most of the delta range. In contrast, IS fell between SWS and REM in the delta band. IS was significantly lower than SWS in the band 1.33–2.67 Hz and significantly exceeded REM between 2 and 8.67 Hz, which includes both delta and theta power. REM had significantly more power in very low frequencies than both SWS (in the range 0–0.67 Hz) and IS (0–1 Hz), which we speculate is due to residual eye movement artifact we were not able to filter out: during analysis, we noted that power <1 Hz fell when removing epochs with large amounts of eye movement artifact, and Wake had high power in this band as well as REM (Fig 4B and 4C). For this reason, we define the delta band as 1–4 Hz in later analyses.

REM contained more high-frequency power than other sleep states (Fig 4B and 4C): it exceeded IS in the range 17.66–55 Hz and exceeded SWS in the range 16.33–55 Hz. IS was intermediate, trending higher than SWS in the range 4–44 Hz. Across all bands, REM was very similar to Wake (Fig 4A–4C; see S2 Fig panel a for variance of Wake spectra). Qualitatively, REM appeared to have more gamma than Wake, but this was not significant. The only statistical difference was a trend for greater power during Wake in the range 4.66–10.33 Hz (Fig 4C, bottom). Interestingly, this partially encompasses the alpha frequency band (8–12 Hz), which is a noted marker of resting wakefulness in humans and one of the key distinctions between REM and wake EEG. To our knowledge, such a rhythm has not been described in any bird species.

We also examined drowsiness, which appeared very similar to quiet wake (S2 Fig panel a) and unihemispheric sleep (S2 Fig panels b-c). During unihemispheric sleep, the awake hemisphere most closely resembled drowsiness (S2 Fig panel b), while the asleep hemisphere was most similar to IS (S2 Fig panel c).

We next examined how the spectral characteristics of sleep changed over time. Slow wave activity (SWA), or the average power in the delta band (here defined as 1–4 Hz) is generally thought to indicate the depth or intensity of NREM. In mammals it is used as a marker of sleep pressure, with prolonged wake leading to higher subsequent SWA. We calculated SWA during NREM for each hour across the 24-h period and compared this to hourly TST (Fig 4D). (Hourly SWA calculated during all sleep yielded essentially identical values.) At night (grey shading) while TST stayed consistently high, SWA started high and decreased thereafter (regression with hour of night, slope = −0.044, $r^2$ = 0.786, $p$ = 0.0003). During the day, SWA started out very low and increased over the first 6 h, concomitant with a prolonged period of wakefulness. Notably, the drops in SWA in hours 7–8 occur during the afternoon nap, visible as a small bump in the TST plot (Fig 4D). SWA again increased over the evening, during another period of relative wakefulness.

The gamma/delta ratio (S1 Data) had the inverse pattern to SWA, as expected (regression with hour of night, slope = 0.060, $r^2$ = 0.87, $p$ = $2.9 \times 10^{-5}$). To determine whether this was due

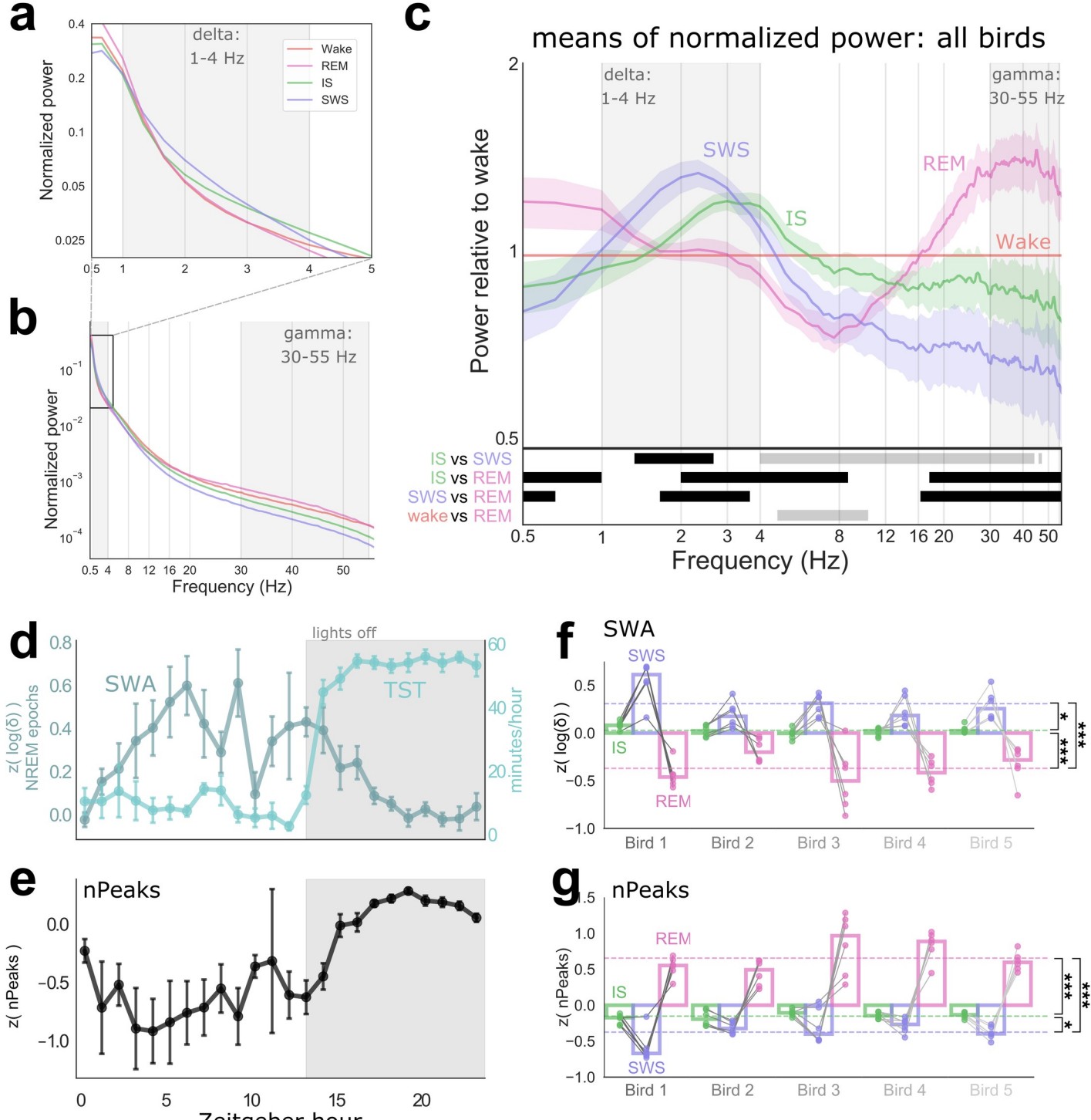

**Fig 4. Spectral characteristics of sleep states.** Average of normalized power spectra of all birds highlighting delta (a) and gamma (b) bands. (c) Top, the 3 major sleep states (averaged normalized power spectra from all birds) plotted relative to Wake (i.e., divided by Wake spectrum). Shading around each line indicates SEMs. Gray boxes highlight delta and gamma frequency bands. SWS had a significant peak in the delta band, while REM peaked in the gamma band. Bottom, statistical comparisons between pairs of normalized spectra (i.e., before spectra were divided by Wake). Black indicates significant differences (Rüger's areas; see text); gray indicates trends (consecutive bins with $p < 0.05$). (d–e) Sleep characteristics by hour, across the 24-h period. Shaded area indicates lights off. Error bars, SEM. (d) Dark blue (left y-axis), z-scored SWA (or delta content; see text) of NREM epochs. Light blue (right y-axis), TST per hour. SWA decreases across periods of higher TST. (e) The nPeaks (z-scored) of all sleep epochs. At night, nPeaks increased concomitant with the amount of REM. (f–g) Characteristics of sleep stages across birds. For each bird, all 6 EEG channels are shown as dots. Bars indicate the median across channels. Dashed lines, means across all birds. Paired $t$ tests: $^*p < 0.05$, $^{***}p < 0.005$. (f)

SWA. (g) nPeaks. Data are provided in S1 Data. IS, intermediate sleep; nPeaks, number of peaks per epoch; NREM, non-REM; REM, rapid eye movement sleep; SWA, slow wave activity; SWS, slow wave sleep; TST, total sleep time.

to changes in delta, gamma, or both, we also examined the number of peaks per epoch (nPeaks), a simple measure of high-frequency activity [53]. In human EEG, the nPeaks of the global field potential is highest during wake and has been shown to decrease with increasing sleep depth [54]. During the day, when REM was low, the nPeaks of sleep epochs stayed low but was extremely variable (Fig 4E). At night, nPeaks started low and increased across the night (Fig 4E; slope = 0.059, $r^2$ = 0.490, $p$ = 0.02), echoing the pattern of REM (see Fig 2D). This indicates that the changes in gamma/delta was likely to have been driven to a substantial degree by changes in gamma.

SWA and nPeaks reliably differed across sleep stages, within all 5 birds and within almost all individual EEG channels (Fig 4F and 4G; SWA one-way ANOVA, F(4) = 35.7, $p$ = 8.9 × 10$^{-6}$; nPeaks, F(4) = 71.7, $p$ = 2.1 × 10$^{-7}$). Gamma/delta (S1 Data) exhibited a pattern very similar to nPeaks (F(4) = 43.3, $p$ = 3.3 × 10$^{-6}$). As expected, individual SWS epochs tended to contain high delta, a low gamma/delta ratio, and low nPeaks/s; REM epochs occurred during periods of low delta, high gamma/delta, and high nPeaks/s; and IS epochs fell in between.

We conclude that budgerigars exhibit extensive REM episodes characterized by a prevalence of high-frequency activity. Spectral analyses support our manual classification of mixed-frequency IS interspersed with low-frequency SWS.

## Ultradian rhythms and sleep cycles

We also observed clear evidence of ultradian rhythms in the distribution of sleep states. Because budgerigar sleep stages alternate rapidly, we examined individual 1-s epochs of EEG during sleep. Delta, gamma/delta, and nPeaks exhibited oscillations throughout the night (Fig 5A) that corresponded to manual sleep scores. Closer inspection revealed that these oscillations corresponded to a reliable alteration between SWS-dominant and REM-dominant sleep (Fig 5B).

We examined oscillations in the gamma/delta ratio at several timescales (Fig 6A and 6B). The 1-min moving average did not identify a rhythm that was consistent across channels or birds (S1 Data). However, the 10-min and 10-s moving averages revealed 2 rhythms with a highly consistent period across birds: an approximately 30-min rhythm (28.66 min ± 4.23, ranging from 24 min to 34 min) (Fig 6C) and a 60-s rhythm (60.37 s ± 13.71, ranging from 41 s to 79 s across individuals) (Fig 6D).

As can be seen in the examples in Fig 6B (also Fig 5), sleep stages varied in a predictable manner with these ultradian rhythms (Fig 6E–6G). This was especially evident for the slow rhythm (Fig 6E and 6F). When the delta/gamma data were fitted with a sinusoid (Fig 6E), the sleep stage fluctuated in tandem with the phase of the slow rhythm: IS (S1 Data) stayed relatively constant over all phases, while SWS and REM occurred approximately 180˚ out of phase. Epochs of REM tended to occur closer to the peak (phase = π/2), while SWS occurred closer to the trough (phase = —π/2) (Fig 6F; one-way ANOVA, F(4) = 24.72, $p$ = 5.6 × 10$^{-5}$). The fast rhythm had a more variable relationship to sleep stage but in general tended to separate REM from NREM (Fig 6G; one-way ANOVA, F(4) = 7.19, $p$ = 0.009).

We also found that the period of the fast rhythm was strongly positively correlated with the amount of REM (S3 Fig panel a; slope = 0.73% REM/s, $r^2$ = 0.943, $p$ = 0.006). Conversely, the period of the slow rhythm had a nonsignificant negative correlation with percent REM (S3 Fig panel b; slope = −1.74% REM/min, $r^2$ = 0.512, $p$ = 0.17). Additional experiments would be needed to confirm these results.

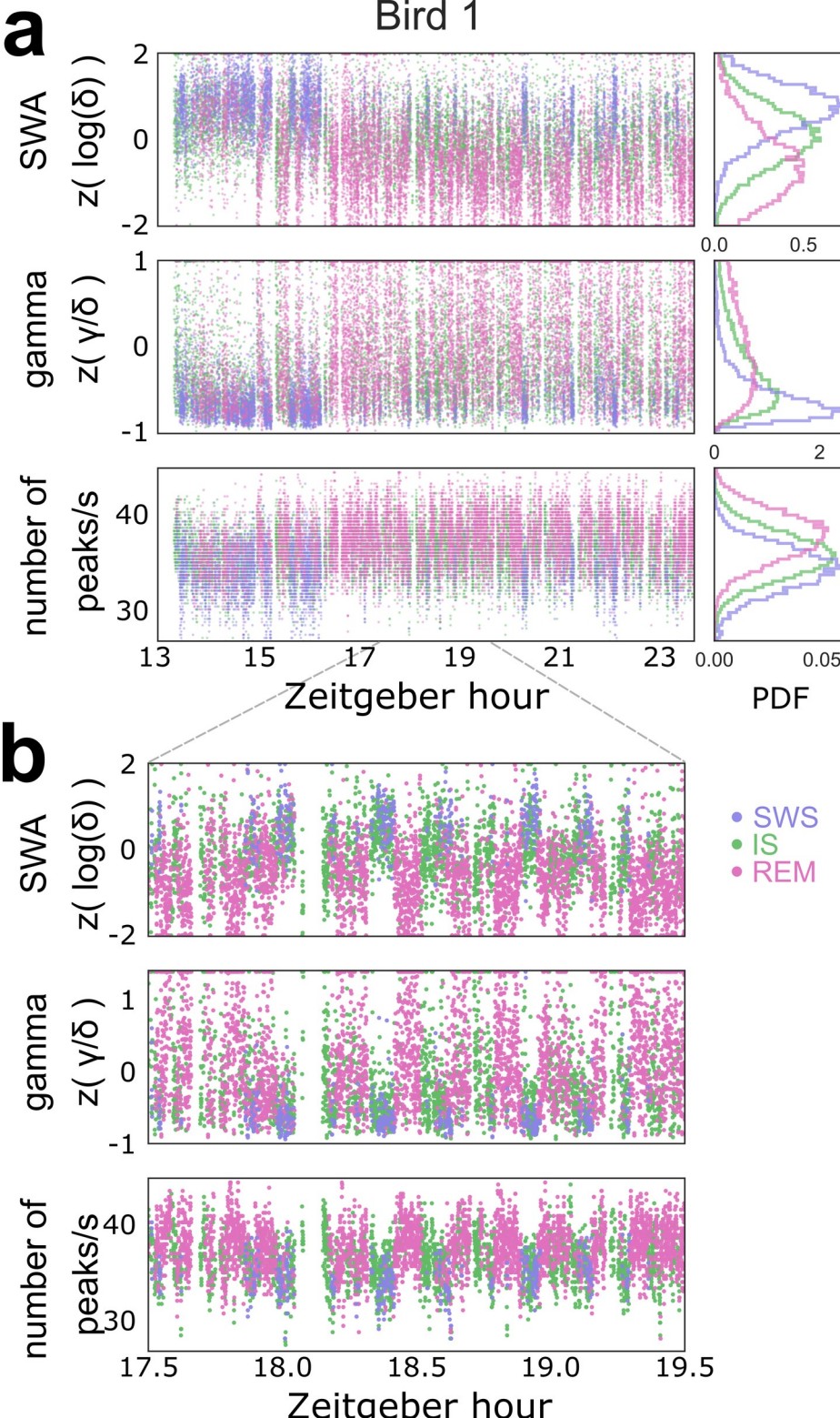

**Fig 5. Coordinated ultradian oscillations in sleep characteristics.** (a) An example of time-varying spectral characteristics of single-channel EEG data across the night. Dots represent epochs spaced 1 s apart. Colors indicate sleep stage. Top: z-scored log of delta per epoch. Middle: z-scored gamma/delta ratio per epoch. Bottom: number of peaks per second, a measure of the prevailing high-frequency rhythm. Right: histograms depicting the probability

density function of the data shown. (b) Two hours of the data shown in (a). Delta, gamma, and nPeaks in this channel oscillate with a period of approximately 15 min. The gap of whitespace just after hour 18 corresponds to a wake episode, which occurs at the end of a prolonged period of SWS-dominant sleep. EEG, electroencephalogram; IS, intermediate sleep; REM, rapid eye movement sleep; SWA, slow wave activity; SWS, slow wave sleep.

Overall, these data suggest that budgerigars undergo both a fast ultradian rhythm similar to that reported in some other animals [55–57] and a slow ultradian rhythm on the order of the human 90-min sleep cycle.

## Slow waves and eye movements

To extend manual sleep scores to unbiased characterization of a larger dataset, we carried out automated detection of slow waves, eye movements, and eye movement artifacts.

Slow wave detection were detected using an adapted version of the zero-crossing method (Fig 7A) [58–60]. As expected, the slow waves occupied the highest amount of time during SWS epochs and the lowest during REM epochs (Fig 7B). The mean seconds/epoch of slow waves identified with this method typically fell above 50% of the 3-s epoch, the criterion for manual scoring of SWS (Fig 7C).

Eye movements were detected as the anticorrelation of the left and right EOGs (Fig 7D). The 2 NREM stages differed from REM in eye movement content (Fig 7E and 7F): In all birds, both SWS and IS epochs were heavily skewed toward 0, while REM epochs formed a much broader distribution with a large peak near 0 but also a long tail extending beyond 2 s. This may indicate 2 subtypes of REM that correspond to tonic REM (no eye movements) and phasic REM. In summary, automated slow wave and eye movement analyses supported our manual scoring and further highlighted a preponderance of REM, both tonic and phasic, in budgerigars.

## Automated scoring of sleep

To further validate our scoring, we performed automated classification of sleep using the epoch-by-epoch spectral characteristics calculated above. We adapted and expanded on an algorithm previously developed for zebra finch sleep [28]. Briefly, this procedure used 2 independent steps of k-means clustering to split epochs into (1) either SWS or non-SWS and (2) either REM or NREM. Both clustering steps were applied to the gradient of delta, gradient of gamma/delta, standard deviation of the waveform, and nPeaks. Step 1 additionally incorporated the variables log(delta) and the absolute maximum amplitude, while step 2 incorporated the gamma/delta ratio. Before clustering, non-sleep epochs and epochs with high-amplitude artifacts were removed and variables were z-scored.

Epochs classified as non-SWS and NREM were scored as IS, while epochs classified as both SWS and REM were designated as artifacts. Scores were then smoothed using a 5-s rolling mean. Prior to score smoothing, the total percentage of epochs scored as artifact was very low, only 1.22% ± 0.76%. Following smoothing, this value was near 0.

Comparisons between the manual and automated scoring showed broad agreement between the patterns of sleep scores across the night (Fig 8A). The automated scores for most birds, if anything, tended to emphasize the nighttime REM increase and SWS decrease, as can be seen in Fig 8A.

Epochs of the automatically scored sleep stages formed visible clusters in the multidimensional space of the spectral variables. When projected onto the three-dimensional space of delta, gamma/delta, and gradient(gamma/delta), SWS stretched out into a "spear" with IS at the base and REM in a cloud behind (Fig 8B and 8C, left). When projected onto the space

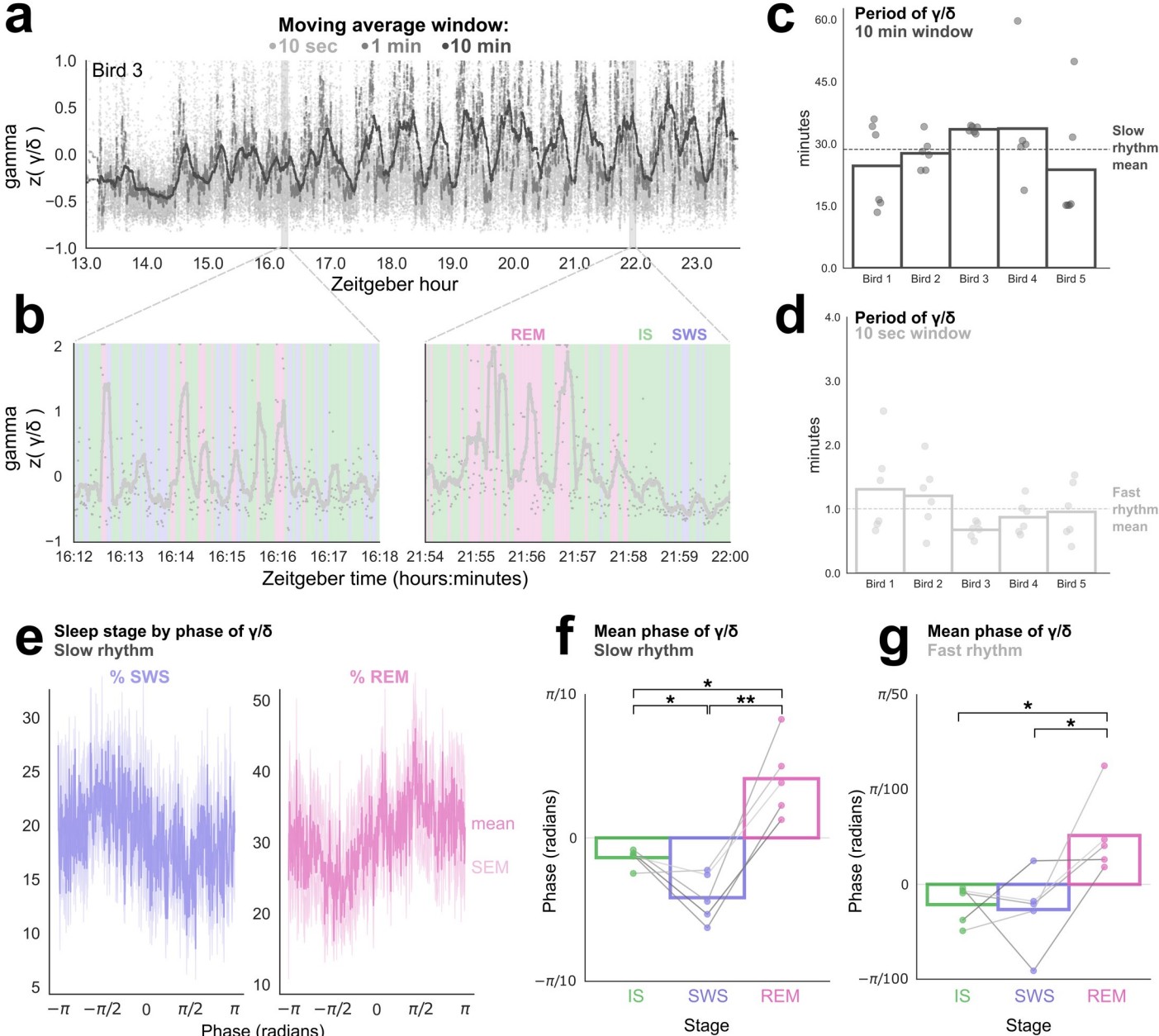

**Fig 6. Characterizing ultradian rhythms in sleep.** (a) The gamma/delta ratio across the night (Bird 3). Moving averages with 3 different window lengths revealed underlying oscillations in the gamma/delta ratio. The slow rhythm cycles approximately every 30 min. Shaded areas correspond to 6-min windows displayed in (b). Each dot indicates one 1-s epoch. (b) The gamma/delta ratio in two 6-min windows as indicated in (a). Very light grey dots correspond to raw data. The solid grey line shows the 10-s moving average. This bird's fast rhythm had a period of approximately 40 s, or 9 cycles per 6 min. Note the tendency for different sleep cycles (colors) to segregate at different phases of the fast cycle. (c) The period for the gamma/delta ratio of the 10-min moving average. Each dot represents a single EEG channel. Each bar represents an individual bird. Mean of birds (dashed line) was 28.66 min ± 4.2 min. (d) Same as (c) for 10-s moving average; mean = 60.37 s ± 13.71. (e) The percent of SWS epochs (left) and percent of REM epochs (right) at each phase of the slow rhythm. The data were fitted to a sinusoid with a trough at -π/2 and a peak at π/2. Dark line shows the mean across birds; light shading indicates the SEM. (f–g) The average phase at which each epoch of a given stage occurs for the slow rhythm (f) and the fast rhythm (g) in the gamma/delta ratio. Each dot represents an individual bird. Each bar indicates the mean across all birds. Lines are colored according to bird identity, from Bird 1 (darkest lines) to Bird 5 (lightest). $^*p < 0.05$, $^{**}p < 0.01$, paired $t$ tests. Data are provided in S1 Data. EEG, electroencephalogram; IS, intermediate sleep; REM, rapid eye movement sleep; SWS, slow wave sleep.

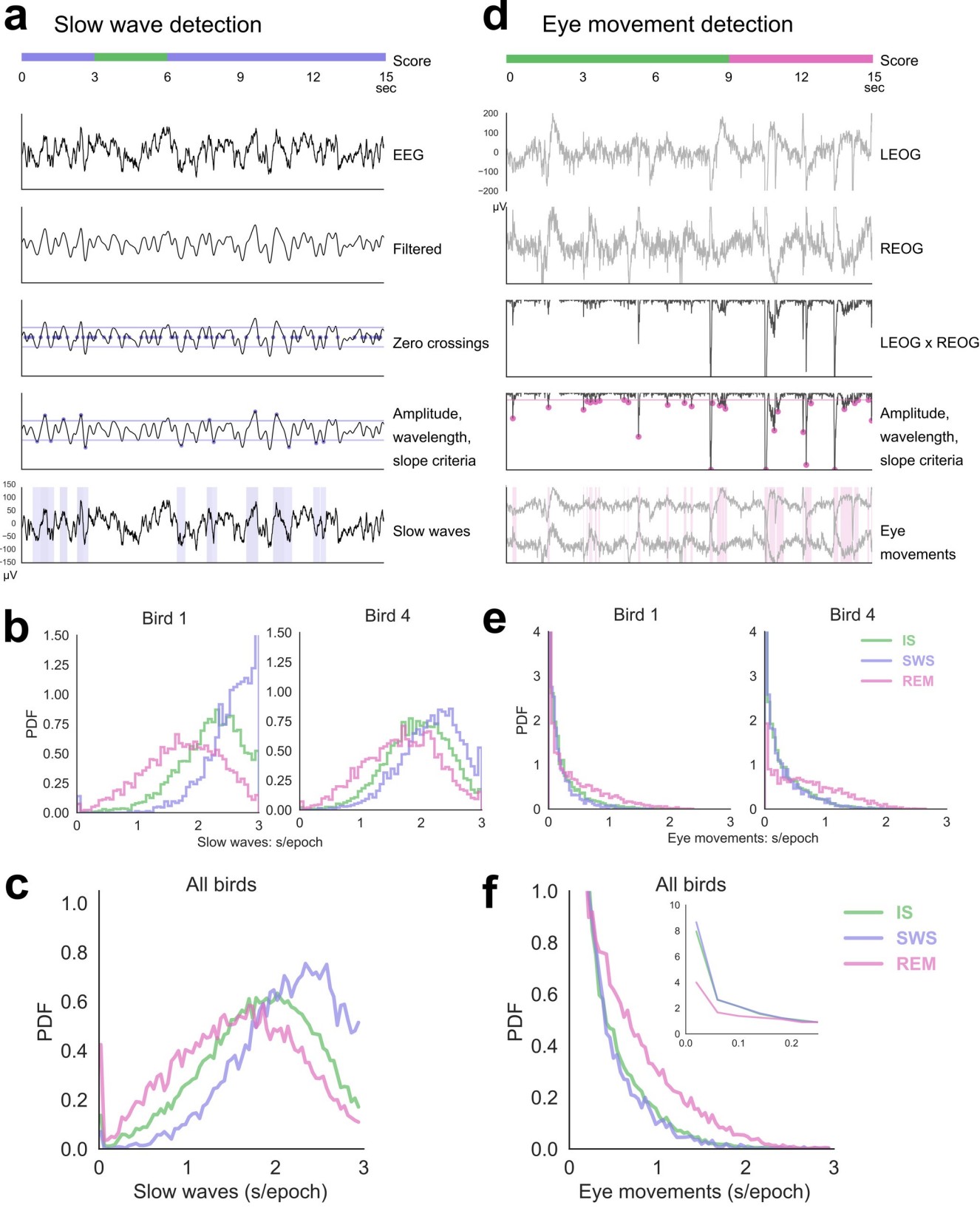

**Fig 7. Automated detection of slow waves, eye movements, and eye movement artifacts.** (a) An example of slow wave detection. The EEG is filtered between 0.5 and 4 Hz, zero crossings are identified, and each resulting half-wave is evaluated for amplitude, wavelength, and slope criteria (see Methods). Eye movement artifacts were also excluded from the pool of candidate slow waves (see text). (b) In 2 example birds, the probability distribution of slow waves per epoch in each sleep stage. Bird 1 has an above-average peak of slow waves/epoch at 3 s. (c) Average of slow waves per epoch across all birds. (d) Example of eye movement detection. Eye movements appear as opposing deflections in the 2 EOG channels. The left EOG is multiplied with the right EOG, and large negative half-waves in this correlation are identified as eye movements. Eye movement artifacts in the EEG (see Methods) were detected with a similar process applied to the EEG × EOG product. (e) In 2 representative birds, the normalized distribution of the number of seconds of eye movements per epoch, for each of the stages of sleep. (f) Average of eye movements per epoch across all birds. Inset, close up of distributions at zero eye movements/epoch. The number of zero-eye-movement epochs during SWS and IS greatly exceeded those during REM. Data are provided in S1 Data. EEG, electroencephalogram; EOG, electrooculogram; IS, intermediate sleep; PDF, probability density function; REM, rapid eye movement sleep; SWS, slow wave sleep.

formed by delta, gamma/delta, and gradient(delta), REM collapsed out into a "spear" with SWS forming a cloud (Fig 8B and 8C, right). There were many epochs identified as IS that did not collapse in either projection (green clouds of points, Fig 8B and 8C).

We also performed principal component analysis on the 5 variables log(delta), gamma/delta, gradient of delta, gradient of gamma/delta, and standard deviation (Fig 8D and 8E). In the space made up by the first 3 principal components, REM and SWS formed 2 orthogonal planes. IS fell into the warped region that linked these planes. These patterns, seen in all 5 birds (S4 Fig), were extremely similar to those found in zebra finch sleep EEG [28].

The overall amounts of each sleep stage found by the algorithm were similar to those obtained by manual scoring across the 24-h period. The proportion of automatically scored REM was 28.6% ± 9.7%, nearly identical to the manually scored value. Compared to manual scoring, the algorithm tended to score slightly less IS (43.2% ± 14.3%, mean ± SD) and slightly more SWS (28.2% of TST ± 15.3%).

The concordance between manual and automated scoring for REM (i.e., considering REM/NREM categories) was 77.57% ± 3.10% (mean ± SEM), comparable if somewhat lower than the 84.30% ± 3.81% previously reported for REM/NREM categories in zebra finches [28]. The concordance we achieved for SWS scoring (hence also recognizing IS) was 71.77% ± 2.29%, which is quite good considering that the original algorithm was developed based on manual REM/NREM scoring only.

To further investigate the performance of the algorithm, we also calculated Cohen's kappa, $\kappa$ (that ranges from 0 to 1), a measure of interrater reliability that controls for chance agreement and has been previously used in human sleep scoring [61]. Kappa for REM scoring was $0.45 \pm 0.04$ (mean ± SEM), considered "moderate agreement" ($0.41 \leq \kappa \leq 0.60$) according to standard benchmarks for kappa [62]. Kappa for SWS scoring was $0.22 \pm 0.06$, and for overall SWS/IS/REM scoring was $0.27 \pm 0.03$, both of these falling into the category of "fair agreement" ($0.21 \leq \kappa \leq 0.40$).

## Effects of constant light

We sought to understand why many observations we report here had been missed in prior studies, and why the amounts of TST and REM we observed greatly exceeded that found previously. In many early sleep studies of birds, direct observation of behavior in the dark was not possible with available technology, especially the inaccessibility of technology for infrared observation. In the case of both previous parrot studies, including a budgerigar study, birds were placed in constant light (LL) and given 7 d to acclimate prior to recording [23,24]. We predicted that LL was responsible for the principal differences between our results and the prior results. To test this, we replicated the conditions of the prior study with 3 of our birds (2 males, 1 female). After several days of baseline (light/dark [LD]) PSG and video recordings, we exposed birds to LL conditions for 8 d, then returned them to LD while continuing to record (see Methods).

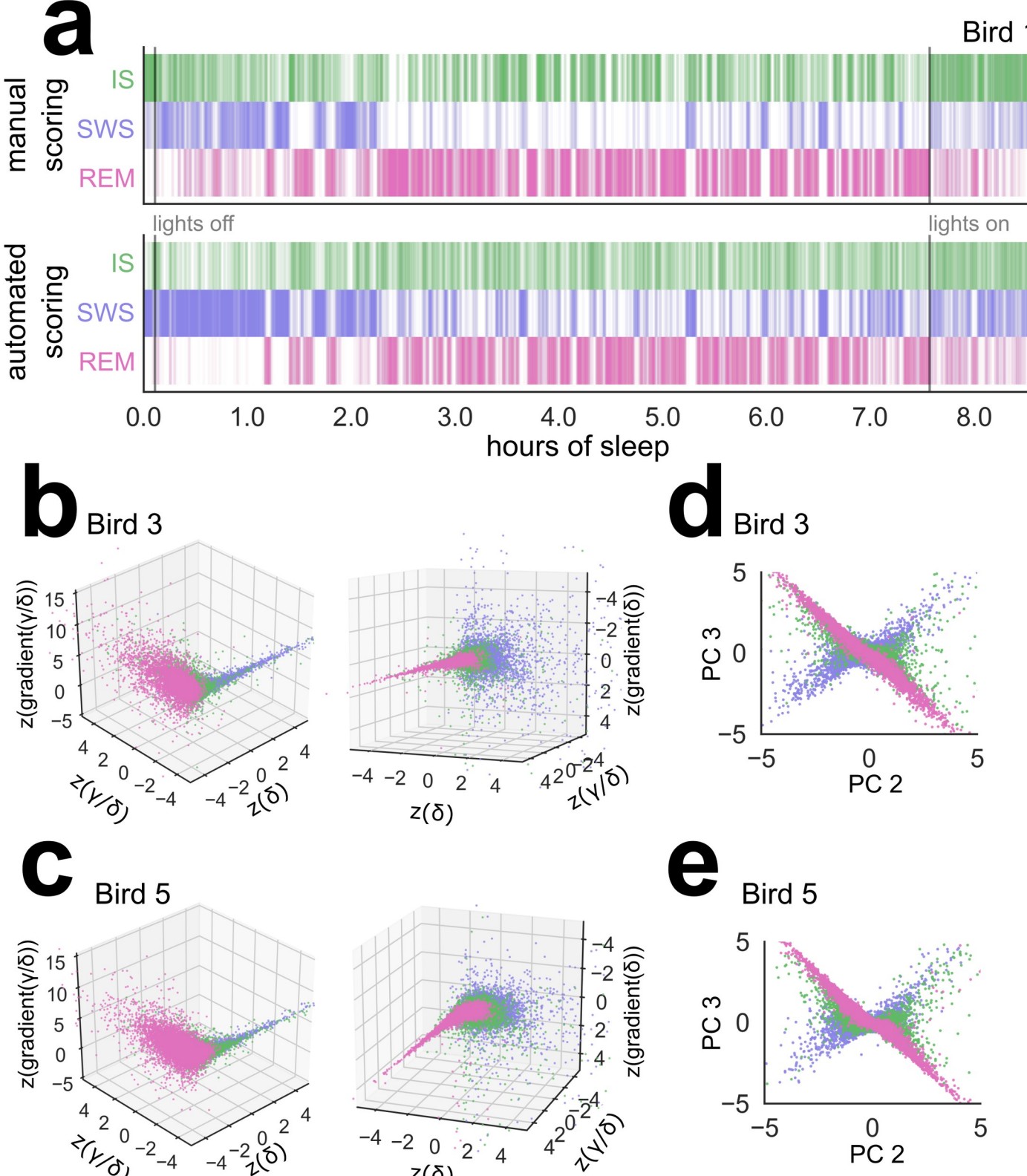

**Fig 8. Automated sleep scoring.** (a) Example hypnogram comparing manual scoring (top) to automated scoring (bottom). Each tick represents one 3-s epoch. There is general concordance between the two (see text). In this example the automated scoring finds somewhat more SWS and less IS and REM. (b) Spectral

characteristics of automatically scored epochs for Bird 3, plotted in a three-dimensional space as in Low and colleagues (2008) [28] (Fig 1C and 1D). Each dot corresponds to one epoch. Colors represent the automatically scored sleep stage as indicated in (a). When the Z-axis is the gradient of gamma/delta (left plot), SWS collapses to a spear; when the Z-axis is the gradient of delta (right plot), REM collapses to a spear. IS includes intermediate points that do not collapse. (c) Bird 5, same as b. (d–e) Results of principal component analysis of automatically scored epochs, plotted as in Low and colleagues (2008) [28] (Fig 1E). The stages of REM and SWS form orthogonal planes, with IS forming the transitional area in between. The first 3 principal components from all birds shown in S4 Fig. Data are provided in S1 Data. IS, intermediate sleep; PC, principal component; REM, rapid eye movement sleep; SWS, slow wave sleep.

We first examined TST as quantified by video behavioral scoring. In baseline conditions, video sleep/wake classification showed good agreement with subsequent PSG scoring (compare Fig 9A top, with Fig 2A). On average across birds, 2.88% ± 0.57% of epochs originally marked as drowsy during video scoring were re-scored as sleep during manual PSG scoring. In a similar fashion, behavioral sleep epochs were sometimes reclassified as drowsy (2.64% ± 0.51% of epochs originally marked as sleep). TST scored by video was 47.09% ± 2.71%, while TST scored by PSG was 47.35% ± 2.67%. The average absolute difference in percentage TST between video and PSG scoring was 1.45% ± 0.38%. In Birds 1, 2, and 4 this difference was an increase (video < PSG), while in Birds 3 and 5 this was a decrease (video > PSG). To facilitate comparison between LD and LL, all TST values reported below are derived from video scores.

Behavior after 7 d of LL was highly disrupted relative to baseline light-dark (LD) conditions in all 3 birds (Figs 9A, 9B and S5). These 3 birds had 26.5% ± 9.97% TST in LL, compared to 46.1% ± 6.91% TST in baseline LD. The amount of sleep was therefore almost halved in LL: on average across birds, the LL:LD ratio of TST was 0.555 ± 0.139. The change in TST both in direction and magnitude is consistent with the previous report of TST in budgerigars (25.15% of time) conducted in LL conditions [23] as compared with our study (47.35% of time, baseline PSG-determined TST in all 5 birds) conducted in LD conditions.

Furthermore, birds in LL almost never engaged in the climbing behavior that they exhibited in the dark period. Instead, they slept on the ground or a low perch, similar to LD daytime napping. This is further evidence that their sleep behavior was disrupted by LL conditions.

Constant light also caused sleep fragmentation. Across 24 h in LD, the 3 birds exhibited 321.3 ± 116 total sleep episodes, whereas in LL this went up to 970.0 ± 381 sleep episodes. The average duration of the birds' sleep episodes in LD was 132.0 s ± 32.4, but in LL this dropped to 25.5 s ± 9.4 ($t(2) = 4.00$, $p = 0.057$). This lack of consolidated sleep periods likely predisposed the birds to a decrease in REM, which tends to occur later in sleep periods. In normal LD, the duration of sleep episodes was significantly longer than the REM latency ($t(4) = 3.43$, $p = 0.026$) (Fig 9C). Indeed, the average REM latency (time between sleep onset and the first REM epoch) of these 3 birds in LD was 43.1 s ± 24.6, which was not significantly different from the duration of sleep episodes in LL ($t(2) = -0.78$, $p = 0.519$). This suggests that sleep was fragmented into episodes that were, on average, too short for birds to enter REM.

Massive disruption of sleep/wake behavior was confirmed in all 3 birds by automated motion detection of continuous video over baseline LD, 8–12 d of LL, and subsequent recovery in LD (Fig 9D and 9E). The response to constant light was somewhat individualized; for example, one bird (Bird 3) showed signs of free-running during the first few days of LL; Bird 4 quickly adopted a fragmented, aperiodic motor rhythm; and Bird 2 exhibited relative inactivity for the first 48 h, followed by increasing hyperactivity (Fig 9D). By day 8, which corresponds to the day of sleep recording in prior studies, all 3 birds appeared arrhythmic (S5 Fig). This was confirmed by periodogram analysis, showing a large peak in the 24-h rhythm in LD that disappeared in LL (Fig 9E). This is consistent with the prior budgerigar sleep study in LL, which reported that, although sleep was increased significantly during subjective night, budgerigars were "polyphasic. . .. [S]leep episodes were distributed irregularly throughout the 24-h period without showing a clear periodicity" [23].

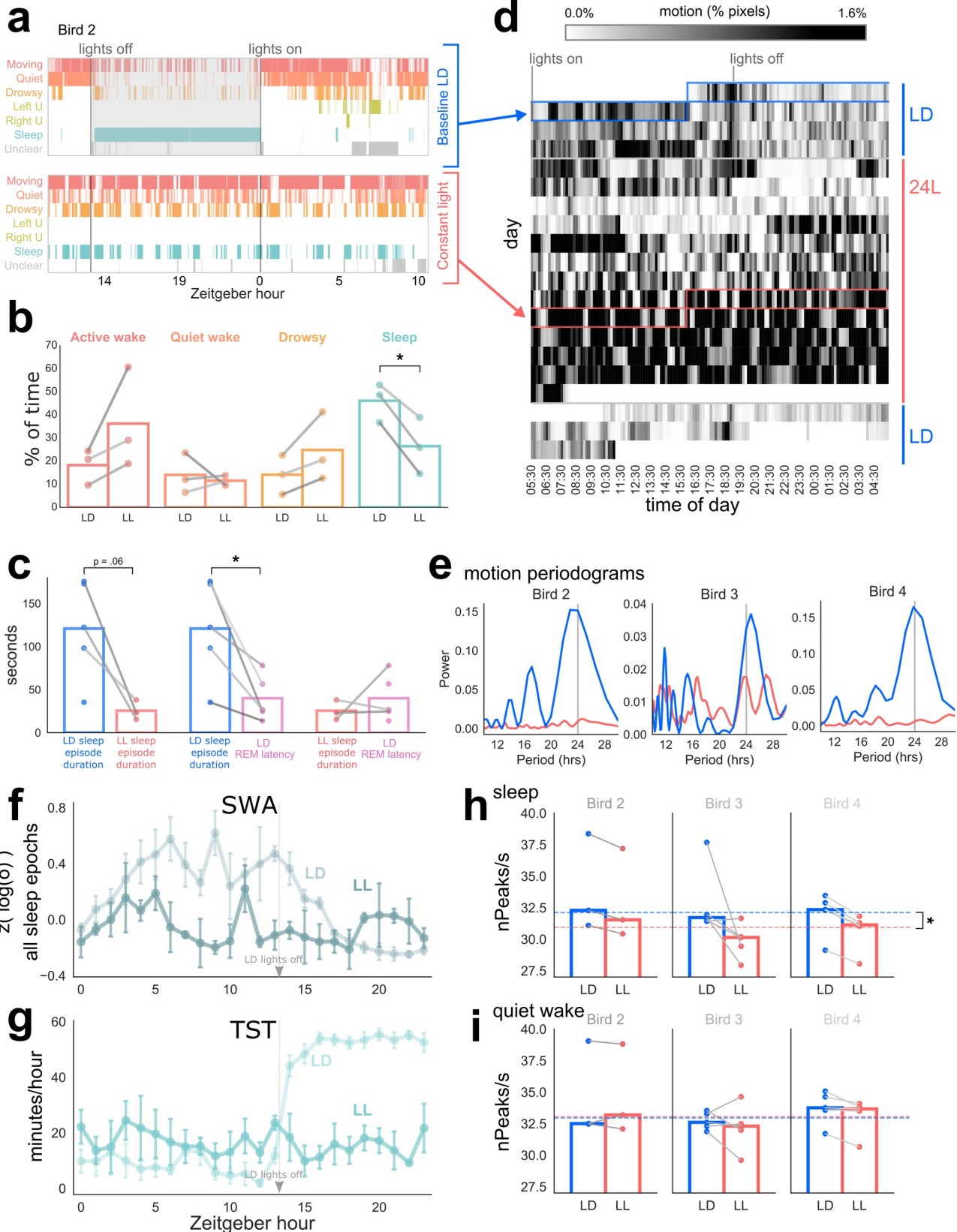

**Fig 9. The effect of constant light on sleep/wake behavior.** (a) Video scores of one bird (Bird 2) across 24 h. Each tick represents one 3-s epoch with rows corresponding to behavioral states (U, possible unihemispheric sleep). Lights off is shaded in grey. Top: baseline LD cycle. Bottom: constant light (LL), day 8. Note the extensive interdigitation of sleep and wake in the LL condition. (b) Proportion of video-scored behavioral states in baseline LD versus day 8 of LL. Each dot represents one bird. Shading of gray lines in b–c corresponds to bird identity as in Fig 2. Note significant reduction of sleep in LL condition (see text). (c) Sleep episode durations in LD (blue) and LL (red), compared to mean REM latency in LD (purple). Note that average REM latency (the average time to beginning of REM following a period of sleep) is shorter than the average sleep duration period in LD but not LL. (d) Actigram of motion detected from video spanning baseline LD conditions, constant light, and return to baseline LD. Each row represents 24 h. Each vertical segment in a row represents average motion in a 10-min bin, where darker shading indicates more motion. Boxed sections indicate days plotted in (a). White area at the end of the LL period corresponding to missing data. (e) Lomb-Scargle periodograms of motion data during constant light (red) and during baseline LD (blue) for each bird. Note large peak at 24-h period (vertical grey line) indicating the circadian rhythm of LD birds, with little or no consistent circadian rhythm for LL birds. (f–g) Loss of circadian rhythm in delta power (f) and TST (g) in LL (darker lines; hourly means ± SEM) compared to LD (lighter lines; hourly means ± SEM; also see Fig 4D and 4E). SWA shown here is for all sleep epochs (rather than NREM only) to allow comparison between LL and LD. (h–i) Mean number of peaks/second in LD versus LL for sleep epochs (h) and quite wake epochs (i). Note significant changes in sleep for LL versus LD but no changes for quiet wake. Each dot corresponds to one EEG channel. Only channels that remained intact throughout both LD and LL are included. Bar plots show the median across channels for each bird. Dashed lines, means across all birds. $^*p < 0.05$, paired $t$ tests. Data are provided in S1 Data. Raw video scores are provided in S2 Data. EEG, electroencephalogram; LD, light/dark; Left U, left-brain unihemispheric sleep; LL, constant light; nPeaks, number of peaks per epoch; NREM, non-REM; REM, rapid eye movement sleep; Right U, right-brain unihemispheric sleep; SWA, slow wave activity; TST, total sleep time.

Sleep fragmentation also greatly increased the difficulty of already laborious manual scoring. For this reason, we chose to analyze PSG activity in LL with automated methods including epoch-by-epoch spectral analysis.

Across the 24 h of LL day 8, there was no consistent pattern in SWA, either overall (Fig 9F; also compare to Fig 4F) or for any of the individual birds. This matched the pattern of hourly TST, which remained at an overall low baseline level with a few variable fluctuations (Fig 9G; also compare to Fig 4G). There was no evidence of a SWA decrease.

We also examined nPeaks/s, a measure of high-frequency activity that is highest during REM, during all sleep epochs (Fig 9H). To confirm that EEG channels were stable across LD and LL, we measured nPeaks during quiet wake (Fig 9I). Changes were assessed by a two-way repeated measures ANOVA on the median nPeaks of each bird, with vigilance state (wake or sleep) and lighting condition (LD or LL) as the within-subject factors. There was a significant effect of vigilance state ($F_{(1,2)} = 24.13$, $p = 0.04$) and a significant interaction of vigilance state with lighting condition ($F_{(1,2)} = 193.60$, $p = 0.005$). The effect of lighting condition alone was not significant ($F_{(1,2)} = 4.10$, $p = 0.18$). Across LL and LD combined, nPeaks was significantly higher during wake than sleep (paired $t$ tests, $t(5) = 4.29$, $p = 0.008$), as expected. Furthermore, nPeaks during sleep was lower in LL than in LD (Fig 9H; $t(2) = -4.95$, $p = 0.04$), a decrease seen in all but one of the individual EEG channels examined. This was in contrast to nPeaks during quiet wake (Fig 9I), which did not change from LD to LL ($t(2) = 0.32$, $p = 0.78$). This suggests that LL decreases the amount of REM.

We conclude that constant light has dramatic effects on budgerigar sleep. This supports our hypothesis that using LL conditions has obscured interpretation of avian sleep in the previous study of budgerigars, and likely of other bird species as well.

## Discussion

We have demonstrated that budgerigar sleep exhibits numerous complex characteristics not previously described. Budgerigars have large amounts of REM, and their NREM is partitioned into SWS and IS. These conclusions based on manual scoring were supported by spectral characteristics, by automated detection of slow waves and eye movements, and by automated classification of sleep states. We also observed a SWS decrease, a REM increase, and a 29-min ultradian rhythm over the night, similar to features observed in the human sleep cycle.

## Budgerigars exhibit large amounts of REM

We found that budgerigars spent over 30% of nighttime TST in REM (26.5% of 24-h TST). This amount is higher than that of most mammals [5] and any bird species reported to date [4,28,29]. It is comparable to the 20%–25% REM in humans [63] and to the 26.3% REM in ostriches [29]. This diverges from a prior study of budgerigars [23] that reported about 5% REM (recalculated to exclude drowsiness from TST). Our results also diverge from many older studies of birds in which REM was likely underestimated [64–66].

Eye movement content during REM was distributed broadly, in that REM included a large number of epochs in which no eye movements were detected. This may indicate distinct tonic and phasic REM, categories often applied to mammalian REM that may have functional differences. The eye movements of phasic REM co-occur with ponto-geniculo-occipital waves, which spread throughout the visual system and facilitate sleep consolidation of learning [67]. One study has found similar waves in the avian optic tectum [50], the analogue of the superior colliculus and part of the visual system. Phasic REM may also correspond to periods of more intense dream experience [68,69]. In contrast, tonic REM, not typically described in birds, is a state of heightened environmental processing [70] with long-range synchrony across cortical areas [71].

REM increased across the sleep period, as occurs in mammals with consolidated sleep patterns and in songbirds [26–28,72,73]. These findings sharply contrast the traditional view that avian REM is "rudimentary" [64], makes up about 7% of total sleep, and has no consistent pattern across the night [22,23,74,75]. These differences likely arise from technical limitations of the older studies (see below).

Why might budgerigars have such an abundance of REM? In humans, REM is associated with procedural and emotional memory consolidation [9,76,77]. REM may be important for brain development and learning during early life [78,79], particularly during critical periods [80] and with regards to motor development [49]. More REM appears in juveniles in many species: humans [63], other mammals [81,82], and some birds [46,83]. Comparative analyses of mammalian sleep have found higher REM in more altricial species, which are born in a less developed state [5,52,84]. Thus, an interesting possible explanation for large amounts of REM in budgerigars (and songbirds) is that it supports the persistence of vocal learning abilities into adulthood [85–88] or more generally, their sophisticated social and cognitive abilities [89,90]. Further studies are needed on the interplay of learning and sleep within budgerigars and other advanced bird species.

## IS is analogous to human stage 2

Few avian sleep studies have differentiated NREM into SWS and IS, due to precedent and the lack of spindles [22,23,45,72,91]. Indeed, spindles are a hallmark of human N2. However, N2 is primarily distinguished from SWS by lack of slow wave content, and human SWS frequently contains spindles [44]. Motivated by previous work finding IS in zebra finches in the absence of spindles [28], we adapted human scoring guidelines to distinguish IS from SWS. This was further validated using a modified version of the zebra finch sleep scoring algorithm [28]. Consistent with human PSG scoring, we suggest that avian IS and SWS occupy opposite ends of a spectrum of NREM, with low and high amounts of SWA, respectively. This is in contrast to, for example, the intermediate "transition to REM" sleep state that has been characterized in rats [92].

We found that budgerigars spent approximately 50% of nighttime TST in IS, and this amount remained steady over the sleep period. This closely resembles the amount and time course of IS in zebra finches [28] and of N2 [93]. Like N2, IS was the most common state

giving rise to REM and SWS episodes, with relatively few transitions directly between REM and SWS. A small portion of IS may also be analogous to human NREM stage 1, which makes up about 6% of sleep in a typical young adult [63]. In budgerigars, the most common transitions from wake to sleep occurred via IS, similar to the role of N1.

N2 is associated with procedural memory consolidation [12,94], although much of this work has focused on sleep spindles [95,96]. Given that spindles are propagated from the thalamus to the cortex, and that thalamorecipient areas of the pallium are not superficial structures in birds, this may preclude detecting spindles with surface EEG recordings in birds. It is an open question whether IS in birds shares a similar function despite an apparent absence of spindles.

## Evidence for SWS homeostasis

Budgerigars spent 18.2% of nighttime TST in SWS, similar to the proportion in humans [63]. We found a decrease in both SWS and SWA over the night. We also noted an increase in SWA over periods of relative wakefulness. This pattern appears in mammals [97,98] and reflects the homeostatic regulation of SWS/SWA [99]. This is consistent with a growing number of recent studies in birds. SWA decreased in blackbirds [26,72], white-crowned sparrows [27], zebra finches [28], and starlings [73]. SWA rebounds after sleep deprivation in pigeons [100] and starlings [73]. Exposing one eye to increased visual stimulation caused a local SWA enhancement in the corresponding hemisphere [101]. Thus, our data support the interpretation that like mammals, birds regulate SWS/SWA in a homeostatic manner.

## A long-period sleep cycle

Humans alternate between NREM and REM in a 90-min sleep cycle. Studies in other animals have often identified shorter sleep rhythms, e.g. a 50-s cycle in mice [56] and a 90-s cycle in lizards [55]. There is also some evidence for a 50-s cycle during N2 in humans [56]. Ultradian rhythms in avian sleep architecture have rarely been examined. REM was found to recur with a period of 1.1 min in pigeons [102] and 8.2 min in burrowing owls [103]. Other studies reported aperiodicity of SWA and/or REM [26,104]. We found that budgerigars exhibit a 60-s cycle, similarly to pigeons, lizards, and mice, and a 29-min cycle with remarkably consistent periods across individuals. The repeated alternation between NREM and REM has been proposed as a key mechanism of sleep-dependent memory consolidation [10]. The existence of similar cycles in budgerigars could support similarities in sleep regulation and function.

## Constant light as a confounding factor

There exist only 2 previous studies characterizing sleep architecture in parrot species: one in budgerigars [23] and one in orange-fronted parakeets (*Aratinga canicularis*) [24]. Both found significant differences in parrot sleep as compared with mammals. Those studies examined birds in constant bright light (60- to 100-watt incandescent bulbs). Other early studies of avian sleep employed dim red light [22,104–106] or dim blue light [107] at night, or chose to forgo nighttime behavioral observation, as infrared video was not widely available. Additional confounding factors in many early studies (but not in the prior parrot studies) include the lack of eye movement measures like EOG, possibly resulting in eye movement artifacts being classified as slow waves.

Constant light, especially bright light ($>10$ lux) [108] is well known as a highly disruptive condition [109] that abolishes circadian rhythms in many diurnal species, including sparrows [108,110], finches [111], and pigeons [112]. Consistent with this, we found that in budgerigars, constant light strongly disrupted the sleep/wake cycle, fracturing their normal diurnal

behavior into an arrhythmic pattern. TST was halved in constant light, which fully explains the large discrepancy in TST between this study and the previous study. Sleep was fragmented into episodes slightly shorter than the normal REM latency, and high-frequency activity during sleep was lessened, further suggesting a decrease in REM. The pattern in SWA was also lost, likely due to the lack of consolidated sleep.

This renders the prior results unreliable and likely explains the discrepancies with the current results. Indeed, we found that during the day REM constituted 6.0% of TST (normal LD), similar to the previous report of 5% REM in budgerigars (LL) [23]. Interestingly, orange-fronted parakeets in the same LL conditions had 14.8% REM, a possible underlying species difference [24]. Perhaps under normal LD, orange-fronted parakeets exhibit a great deal of REM.

These observations motivate revisiting phylogenetic analyses of avian sleep, which have thus far found little correlation between sleep architecture and species traits [4]. In contrast, a nearly identical meta-analysis in mammals yielded a wealth of associations, e.g. between REM and encephalization [5]. Our findings suggest that suboptimal experimental conditions may have masked analogous links in birds.

### Evolution of sleep

An emerging body of work points to many similarities across mammalian and avian brains and behavior. Our findings extend recent evidence that avian sleep exhibits more REM, more features, and more complexity than were historically recognized. We confirm complex sleep in a species of parrots (Psittaciformes), the sister taxon of songbirds (Passeriformes). The consensus result in songbirds and parrots indicates strong similarities to the structure of mammalian sleep [28]. Additional similarities between birds and mammals arise considering the role of sleep in learning and memory. Behavioral developmental studies demonstrate a relation between sleep and zebra finch song learning [34,35], studies in adult starlings demonstrate a role of sleep in memory (re)consolidation [37,38], and zebra finch electrophysiological studies demonstrate neuronal bursting [113] and song replay [33,35,36,113] during sleep.

How did these extensive similarities arise in birds and mammals? One hypothesis is that sleep in birds results from convergence of similar traits through independent evolutionary processes [15,114,115]. The difference in sleep structure comparing ostriches [29] and tinamous [116], and recent work in a lizard (*Pogona vitticeps*) [55,57], give question to those predictions, although many more sleep studies in reptiles are required. The present results motivate an alternate hypothesis, that the similarities arose by parallel evolution acting through processes of deep homology; this emphasizes greater similarity of ancestral mechanisms of sleep, shared across some reptilian taxa and including birds and mammals. In this hypothesis, attributes of complex sleep patterns appeared during evolution at multiple loci by modification of these shared mechanisms. How would such deep homology be expressed in brain functional anatomy? Recent studies identify similar cell types and circuits in a canonical forebrain pattern of connections in reptiles, birds, and mammals [117], providing a mechanistic focus for such a hypothesis [118]. It remains unresolved whether such constraints of connectivity could give rise to the similar patterns of behaviors and physiological properties in birds and mammals.

## Methods

### Animals

Adult budgerigars (*M. undulatus*) (*n* = 5 [3 female, 2 male]) were obtained from Magnolia Bird Farm (Riverside, CA). In what follows, Birds 1, 3, and 5 were the female birds. Birds were housed in large group cages in a 13L:11D photoperiod and allowed to acclimate to the lab environment for at least 3 mo before recordings began. Parakeet seed mix, water, and cuttlebone

were provided ad libitum. Starting 1–5 d before surgery, birds were transferred to an individual recording cage in an acoustically and electrically shielded enclosure (18" width, 10" height, 14" depth). We attempted to mitigate the effects of social isolation by (1) housing birds together in groups of 2–5, until a few days before surgery; (2) providing mirrors, which have been shown to provide a degree of social stimulation in budgerigars [90], and (3) opening sound box doors when not recording, especially in the days after surgery (all birds were located in the same room), which allowed birds to interact vocally. Perches were placed close to the floor to avoid tether wrapping. The temperature was maintained at 20°C–22°C.

## Ethics statement

All procedures were conducted in accordance with the Animal Welfare Act and were approved by the Institutional Animal Care and Use Committee at the University of Chicago (protocol ID 56471).

## Surgery

Following transfer to the recording cage, birds were implanted with PSG electrodes, which consisted of 6 EEG electrodes, 3 EOG electrodes, and 2 subcutaneous ground electrodes.

Birds were anesthetized with isoflurane gas, placed in a stereotaxic apparatus, and head-fixed via custom-sized ear bars. After removal of the feathers on the top of the head, the skin was sterilized with iodine, and lidocaine cream was applied. A Y-shaped incision was made in the skin, exposing an area of skull approximately 1 cm in diameter. During this process, blunt dissection was used to create the subcutaneous channels that would later accommodate the EOG electrodes. The upper layer of the skull was removed over both hemispheres.

All PSG electrodes consisted of flattened segments of silver-plated 32 AWG copper wire; the final size of each flat electrode was approximately 1 mm × 2 mm in size. These electrodes typically had an impedance of 2–10 kΩ in vivo.

To implant EEG electrodes, the trabeculae were shaved down over each of the desired sites, and a small incision was made in the lower layer of the skull. The electrode was then positioned parallel to the surface of the head and slipped into the incision to rest between the skull and the dura. Three EEG electrodes were implanted over each hemisphere (approximate coordinates relative to the center of the Y-sinus): a frontal electrode (3 mm lateral, 6 mm rostral), a central electrode (3.5 mm lateral, 2 mm rostral), and a posterior electrode (4 mm lateral, 2 mm caudal). In choosing electrode locations, we prioritized avoiding large blood vessels and spacing the electrodes as far apart from each other as possible. Within each bird, the EEG signals were markedly similar between the 3 electrodes on each hemisphere—the main difference being that eye movement artifacts were more prominent on more frontal channels—suggesting that exact location does not strongly affect EEG recording with this type of electrode.

Three EOG electrodes for measuring eye movements were implanted under the skin: one electrode 3–4 mm lateral to each eye and one at the midline (approximately 6 mm medial to each eye). Each of the lateral EOG electrodes were later referenced to the central EOG, similar to the AASM sleep montage in which lateral EOG electrodes are each referenced to a contralateral mastoid electrode [44].

Ground was connected to 2 electrodes implanted under the posterior skin over each side of the cerebellum. All PSG electrodes were wired to a connector (Omnetics, prewired 18 pin dual row Nanoconnector; Minneapolis, MN). Electrodes and connector were affixed to the skull with dental acrylic reinforced with cyanoacrylate glue.

We chose not to use electromyography (EMG) because muscle atonia during REM, although nearly ubiquitous in mammals, is not a reliable phenomenon in birds [27,66,72,91,102,105,119–

121] and was not reported in either of the previous parrot studies [23,24]. For this reason, EMG is often eschewed in more recent avian sleep studies [47,75] and when measured is used less to differentiate REM from NREM than as an indicator of gross movement during wakefulness [30], especially when video recordings are not possible [29,45,74,119]. Similarly, we did not detect signs of overt or reliable atonia during REM in our initial recordings in budgerigars and therefore chose not to introduce EMG. Indeed, nighttime REM usually proceeded while birds were perched high on the walls of the cage, with no sign of loss of balance.

## Data acquisition

Birds were allowed to recover from surgery for 3–5 d before being fitted with a lightweight cable that attached to a 12-channel mercury commutator (Dragonfly, Ridgeley, WV) on top of the cage. This allowed birds to freely move about the enclosure. After at least 4 d of acclimation to the cable, baseline sleep recordings were collected (9–15 d following initial transfer to the recording cage).

Video from a webcam (Logitech) with its IR filter removed was captured to disk, for the first bird using the software guvcview (MKV file format, 640 × 360 resolution, 10 frames per second), and for the other 4 birds using the software MEncoder (AVI file format, 640 × 480 resolution, 30 frames per second). We found the latter approach more reliable. Several mirrors were placed on the walls to facilitate visualization (especially of both eyes), and an IR light provided illumination during the night. Video recordings were reviewed daily to determine each bird's habitual sleeping location, and the camera was moved accordingly. A sheet of Plexiglas was attached to the cage wall in front of the camera to prevent birds from climbing directly on this wall and obstructing the camera view.

After passing through the commutator, EEG and EOG signals were amplified, bandpass filtered (0.1–200 Hz), digitized at 2,000 Hz with a 16-bit converter, and recorded to disk using an amplifier board (model RHD2132; Intan Technologies, Los Angeles, CA) connected to an RHD2000 USB interface board (Intan). Offline, data were bandpass filtered between 0.5 and 55 Hz, digitally referenced, and down-sampled to 200 Hz. Right and left EOG signal were each referenced to the central EOG electrode. EEG signals were each referenced to more posterior electrodes on the ipsilateral hemisphere, resulting in a total of 3 possible EEG derivations on each hemisphere (frontal-central, frontal-posterior, and central-posterior).

## Constant light

Three budgerigars underwent a constant light manipulation. After 1–3 nights of baseline sleep were collected for Birds 2, 3, and 4 (6, 8, and 10 d after initial tethering, respectively), the light cycle was switched to continuous 24-h light lasting 8 d. This mimics the experimental conditions of a prior study of budgerigar sleep [23].

On days 1, 4, and 8 of constant light, birds were tethered for 24 h starting at or before 16:00 in order to collect PSG. Video recordings were collected throughout the entire period of constant light. The camera was not moved throughout the period of constant light and subsequent recovery.

After 8–12 d of constant light (12, 11, and 8 d for each respective bird), birds were returned to their normal 13L:11D light cycle. PSG and video recordings were continued for 1–3 d of recovery.

## Behavioral video scoring

One continuous 24-h recording was analyzed per bird. Climbing behavior at night resulted in unexpected difficulty in capturing continuous behavior and an adequate view of the birds' eyes

with a single camera, despite the use of mirrors. Nights chosen for scoring were captured after multiple days of experimenting with camera placement. Usually, each bird slept in one of a few preferred locations unique to each individual. This could include a preference to sleep upside down.

Prior to scoring, the video and PSG data were extensively reviewed to establish a scoring system. Some avian species exhibit NREM with their eyes open, e.g., pigeons [100], ostriches [29], and tinamous [30]. We did not find evidence for this in the budgerigars. Although drowsiness often included mixed EEG activity with delta content (0.5–4 Hz), this coincided with wake-like behaviors: slow eye movements, blinking, head movements, and fluffing of the feathers. We concluded that sleep can be determined behaviorally by eye closure, as in most birds [120,121]. Therefore, we proceeded as follows, and later examined epochs of behavioral drowsiness for potential re-classification as sleep (see "Manual PSG sleep scoring" below).

We used a 2-step scoring process to determine vigilance states: behavioral video scoring (Table 3) followed by PSG scoring (Table 4). We choose to use 3-s epochs because sleep states could fluctuate rapidly in the budgerigars, as in most birds, including songbirds [28].

Video recordings of behavior were reviewed, and each epoch was scored as active wake, quiet wake, drowsy, possible unihemispheric sleep, or (bihemispheric) sleep (Table 3). Active wake included any overt movements such as climbing, eating, flying, or grooming. During quiet wake, birds remained alert but became still, with an upright posture and infrequent, quick eyeblinks. As birds became drowsy, they began to blink slowly and close their eyes more frequently and would often lean forward or tilt to one side. During sleep, birds fully closed both eyes, the head and body relaxed into a more horizontal posture (sometimes leaning against a nearby wall), and breathing become slow and deep. Breathing often caused noticeable rhythmic rocking movements that were not apparent when birds were awake. Sleep was not scored unless both eyes were completely closed for the entire epoch. If one eye was observed to be closed and the other open, the epoch was marked as potential unihemispheric sleep. This was mostly observed in otherwise drowsy birds.

The deep breathing activity and concomitant whole-body rocking were particularly reliable indicators of behavioral sleep. This facilitated identifying awakenings, which were typically sudden and accompanied by abrupt movement, fast head shakes, and raising of the head. This

**Table 3. Behavioral scoring criteria.**

| Stage | Posture/behavior: Criteria | Eyes: Criteria | Other notes |
|---|---|---|---|
| Active wake | Overt movements | Eyes fully open with infrequent, fast blinks | Can include climbing, eating, grooming, vocalizing, flying, etc. |
| Quiet wake | No overt movements | | Body held still with an upright posture |
| Drowsy | No overt movements | Frequent slow blinking and partial closure of eyes | • Body may lean forward slowly<br>• Head may tilt back or to one side |
| Possible unihemispheric sleep | Similar to either drowsiness or to full sleep | One eye closed and one eye open | • Difficult to detect, must observe both eyes simultaneously<br>• Typically occurs during long periods of drowsiness<br>• The open eye often appears drowsy with frequent blinking |
| Sleep | • Relaxed body posture<br>• Deep rhythmic breathing, often causing the body to rock slightly | Both eyes closed | • At night, prior to falling asleep budgerigars often climb to a position high on the walls or ceiling of the enclosure<br>• The transition out of sleep is often accompanied by abrupt movement, fast head shaking<br>• Occasional muscle twitches and head drooping<br>• Less frequently, rhythmic movements can occur, e.g., fast beak movements resembling vocalizing |

Criteria used to score sleep stages while viewing video of subjects.

**Table 4. Sleep scoring and vigilance state characteristics.**

|  | EEG | EOG | Transitions |
|---|---|---|---|
| **Wake** | • Active wake: Nearly constant movement artifacts<br>• Quiet wake: "flat" EEG (low-amplitude, high-frequency) approximately 25 µV peak-to-peak | • Frequent eye movements, usually large and rapid | |
| **Drowsy** | • Mixed EEG: Wake-like "flat" activity interspersed with slower, higher-amplitude elements that begin to resemble sleep | • Eye movements not uncommon, often slower than during full wakefulness | • Drowsiness can include sleep-like behavior with a wake-like EEG that immediately precedes or follows clear behavioral wake |
| **Unihemispheric sleep** | • Contralateral to closed eye: slower, higher-amplitude activity<br>• Contralateral to open eye: typically wake-like activity | | |
| **IS** | • Slower and higher amplitude than wake or REM<br>• Less than 50% of epoch contains slow waves<br>• Delta activity (0.5–4 Hz) that does not meet amplitude criteria for SWS<br>• K-complex–like events, resembling a single slow wave, can occur | • Infrequent eye movements; usually slow and low amplitude | • Transitions back and forth between IS and SWS are very frequent, as the SWA of each epoch varies<br>• Transitions from REM: Appearance of delta activity or K-complex with amplitudes exceeding that of typical wake or REM |
| **SWS** | • At least 50% of the epoch must contain slow waves<br>• Slow waves: delta frequency (0.5–4 Hz), amplitude at least 4 times that of typical quiet wake activity<br>• Must rule out eye movement artifacts, which can resemble slow waves and are more prominent in frontal channels | | |
| **REM** | • Low-amplitude, high-frequency EEG similar to wake<br>• Theta waves (4–8 Hz) possible; these can be slightly higher in amplitude than typical wake activity<br>• Eye movement artifacts are common and can have the appearance of slow waves, masking an otherwise "flat" EEG | • Large rapid eye movements are typically observed but are not required<br>• Eye movements often start out small and become larger and more frequent as the REM episode progresses | • Transitioning into REM: Eye movements can appear in the EOG shortly before the EEG transitions to REM-like activity; in this case, the EEG is used to score REM onset |

Criteria used for manual scoring of sleep stages from PSG data.

**Abbreviations:** EEG, electroencephalogram; EOG, electrooculogram; IS, intermediate sleep; PSG, polysomnography; REM, rapid eye movement sleep; SWS, slow wave sleep

distinction was not absolute, however. Birds were sometimes seen to wake up without moving, opening their eyes suddenly and then falling asleep again after a few seconds.

We occasionally saw twitching, eye movements, or drooping of the head; epochs that contained these behaviors were annotated. Muscle twitches were usually of the head and beak, and sometimes involved the wings or the whole body. Twitches often occurred in clusters and appeared involuntary compared to movements during wake; for example, even during particularly violent twitches, sleeping birds often showed no sign of righting themselves despite coming close to falling off their perch. Twitches could sometimes trigger an awakening during which the bird righted itself and changed posture. Most commonly, twitches were small and relatively subtle, but it was not unusual to see larger twitches, sometimes occurring in long trains of repetitive movements. These features of twitches were observed in both sexes.

Eye movements during sleep were visible in some birds but difficult to observe reliably given the variability in sleeping position and location between birds. Likewise, drooping of the

head was sometimes observed, but birds often slept with their head resting against a wall, precluding consistent observation of this behavior.

## Manual PSG sleep scoring

After behavioral scoring, EEG and EOG signals were visualized, and sleep architecture was manually scored using a custom GUI written in Python. All scoring was performed according to American Academy of Sleep Medicine guidelines [44] by an experienced scorer trained in scoring human sleep. Table 4 summarizes criteria used for scoring, along with additional PSG characteristics of each vigilance state. Two randomization/blinding methods were used to rule out order-of-scoring effects. For Bird 3, each hour of the night was scored in a randomized order. For Birds 4 and 5, each hour was scored in a randomized order, and the scorer was blind as to time of night. We found many consistent patterns across all 5 birds, with variation independent of the blinding method, and analyzed the data from all the birds together.

Wake epochs were examined qualitatively but never re-classified as sleep based on the EEG alone. Drowsy epochs were reviewed and occasionally re-classified as sleep after examination of the PSG and video together. This tended to happen immediately following or preceding periods of wake.

Epochs marked as potential unihemispheric sleep was scored as such if the hemisphere contralateral to the closed eye exhibited signs of NREM (see below) while the ipsilateral hemisphere showed signs of wake. If the EEG did not meet these criteria, the epoch was marked as drowsy.

Epochs marked as behavioral sleep were classified as SWS, REM, or IS (Table 4) as follows:

SWS was scored when at least 50% of the epoch contained slow waves on one or more channels. Slow waves were defined as delta waves (0.5–4 Hz) with an amplitude at least 4 times the typical artifact-free waking amplitudes; for most birds, this was approximately 125 μV. For Bird 1, which had unusually high amplitude EEG throughout the recording, a criterion of 250 μV was used. The criterion of 50% slow waves, rather than 20% as in human sleep scoring, was adopted due to the short length of the epochs. Small eye movements could appear in the EOG during this stage but were typically not present. In addition to slow waves, the EEG during NREM sometimes exhibited other low-frequency elements, including artifacts from eye movements and very low frequency baseline fluctuations approximately 0.5 Hz. These artifacts were identifiable as excessively synchronous events across multiple channels.

REM was scored when at least 50% of the epoch consisted of low-amplitude high-frequency EEG similar to wake. This EEG pattern was often, but not always, accompanied by rapid eye movements in the EOG, which were rarely visible on the video. This is distinct from our prior report of sleep staging in zebra finches, where visible eye movements during sleep were commonly observed and in fact were one requirement for scoring REM [28]. However, the budgerigars had a much larger enclosure, chose far more varied sleeping locations and positions, and so were difficult to film optimally (i.e., with both eyes clearly visible). Future studies should employ more specialized video recording techniques if visible eye movements are of interest. We were able to compensate for this with the use of 2 EOG channels (2 lateral electrodes each referenced to a central electrode); the EOG measures the dipole formed by the positively charged cornea and the negatively charged retina. Concordant bilateral eye movements appear as diverging waves in the 2 EOG channels, which helps to distinguish true eye movements from artifacts caused by the eyelids and the nictitating membranes [3,30]. Observations during wake confirmed good correspondence of the EOG channels to eye movements.

Eye movements were not obligatory for defining REM epochs, but they were often detected shortly before or after the onset of REM-like EEG and were used to distinguish borderline

REM from IS. Any occurrences of delta waves marked a transition out of REM, either to SWS or IS. Importantly, eye movements often contaminated the EEG, especially more frontal channels, and could resemble slow waves. Such artifacts would likely complicate the scoring of REM based on EEG alone.

IS was scored when at least 50% of the epoch consisted of NREM without sufficient SWA to meet criteria for SWS. We approached the scoring of IS as analogous to human stages 1 and 2. Spindles were not observed in the EEG, so these could not be used to identify transitions from REM to IS. Instead, these transitions were delineated by the appearance of delta activity that was not of sufficient amplitude or quantity for the epoch to be marked as SWS.

## Spectral analysis

Epochs containing large artifacts (mean ± 4 SD of the entire EEG) were excluded from analysis, as were periods scored as active wake.

**Power spectra.** The channels marked as the highest quality during manual scoring were used for spectral analysis (2 channels per bird, from opposite hemispheres for 3 birds and from the same hemisphere for 2 birds). Epochs with more than 0.25 s of eye movement artifact were also removed (see "Event detection" below). We found that removing eye movement artifact consistently reduced power in the 0–1 Hz band.

Only channels with >20 epochs of artifact-free data of a given stage were included; in practice, all 10 channels met this criterion for all stages except unihemispheric sleep. For the awake hemisphere, 5 channels from 3 birds (Birds 1, 3, and 5) had sufficient data to be included; for the asleep hemisphere, 4 channels from the same 3 birds were included.

Multitaper spectrograms were calculated for each 3-s epoch with no overlap (custom-written Python library Resin [see "Code accessibility" below]; NW = 3, number of tapers 2) across the entire 24-h period and averaged across all epochs of each stage. The resulting spectra had a resolution of 0.33 Hz. Normalization and significance testing were then performed following a procedure previously used on canine PSG [122]. To normalize for differences between channels and birds, each spectrum was divided by the total power between 1 and 55 Hz. This range was chosen due to the large contribution of eye movement artifact to the power below 1 Hz. Rüger's areas were then calculated to identify statistically significant differences between the normalized spectra [122]; this is further described below ("Experimental design and statistical analyses").

**Epoch-by-epoch analysis.** To visualize changes in spectra across time, we averaged the mean activity in the frequency bands of delta (1–4 Hz) and gamma (30–55 Hz) for each 3-s window sliding by 1 s, based on previously published analyses [28]. From these we calculated (1) SWA as the log-transformed delta and (2) the gamma/delta ratio. These values were z-scored across all sleep epochs in order to allow comparisons across channels and birds. We also calculated (3) the number of peaks per second (nPeaks/s) of each epoch. This measure captures the most prominent high-frequency rhythm in a given epoch. Unlike gamma/delta, nPeaks/s is independent of delta power, and we found that, relative to any measures derived from the FFT, it is resilient to many types of artifact. Hourly SWA and nPeaks was calculated by averaging values from the 2 best channels from each bird, as above. Otherwise, epoch-by-epoch measures were examined for each EEG channel separately. The median was taken across channels to yield a value for each bird.

**Constant light.** We also conducted spectral analyses of data from birds in the constant light (LL) condition. Spectral analyses of EEGs were performed only on channels that remained intact throughout both baseline and constant light conditions (3/6 channels for Bird 2 and 5/6 channels for Birds 3–4). These intact channels included the "two best channels" used

to calculate hourly SWA for each bird. To compare epoch-by-epoch nPeaks in LL versus LD across all sleep versus all quiet wake, the median was taken across intact channels to yield a value for each bird.

## Ultradian rhythms

We quantified ultradian rhythms in the gamma/delta ratio using a procedure similar to that used in bearded dragons [55,57]. To capture the different frequencies of oscillations we observed, we calculated the rolling mean (one-sample step size) using 3 different window sizes: 10 s, 1 min, and 10 min. We obtained the autocorrelation at each timescale and took the first nonzero peak as the dominant period of that oscillation. Values of a period were averaged across all channels for a given bird.

To characterize the relationship between ultradian rhythm and sleep stage, we used a simple fitting procedure. A sinusoid was fit to the raw data, using the function optimize.curve_fit (10,000 maximum iterations, method "dogbox") in the SciPy Python library [123]. For each channel, the period calculated by the autocorrelation procedure was used as a starting point. The amplitude and offset were set to constant values (0.25 and 0, respectively). We then obtained the percentages of SWS, REM, and IS for each 0.01-radian step from -π to π. For each epoch of a given sleep stage, the average phase was calculated.

## Event detection: Slow waves, eye movements, and eye movement artifacts

To validate our manual scoring and further characterize budgerigar sleep, we applied several automated analyses to periods of sleep. Slow wave detection was carried out using the zero-crossing method (Fig 7A) [58–60]; EEG signals from periods of behaviorally defined sleep were first filtered in the delta range (0.5–4 Hz). Individual half-waves between 2 adjacent zero-crossings were extracted and rectified. A series of criteria were then applied to determine which half-waves qualified as slow waves. Two inclusion criteria were applied: (1) a wavelength corresponding to delta and (2) an amplitude >75 μV peak to peak (37.5 μV for a half-wave). We applied 3 exclusion criteria: (1) waves with a peak >300 μV were excluded as large-amplitude artifacts; (2) waves with 50% or more overlap with detected eye movement artifacts (see below) were excluded; and (3) waves with multiple peaks were discarded if any of the troughs between peaks were <150 μV; this was done to exclude larger fast waves occurring on a small fluctuating baseline. For each bird, time spent in slow waves was collapsed across channels to obtain the total amount of time during which a slow wave was occurring in any channel.

Eye movements appear in the EOG as large rapid fluctuations with opposite polarities in the left and right channels. To detect these periods of anticorrelation, we took the product of the left and right EOGs and searched for periods in which this correlation crossed a negative amplitude threshold. A zero-crossing analysis similar to the slow wave detection method was applied (Fig 7D). Negative half-waves in the anticorrelation were considered eye movements if they had (1) a peak between −5,000 and −500,000 μV$^2$ in amplitude, (2) a wavelength corresponding to 0.2–60 Hz, and (3) a maximum negative slope in at least the 75th percentile of slopes for that dataset. In one bird (Bird 5), the right EOG was very low amplitude, so the anticorrelation amplitude criterion from step 1 was set to −250 μV$^2$.

Eye movement artifacts for each EEG channel were detected by applying a similar procedure as for detecting eye movements. The product of the EEG channel and its ipsilateral EOG was calculated. Eye movement artifacts were defined as either positive or negative waves of at least 10,000 μV$^2$ in amplitude with (1) a wavelength corresponding to 0.2–60 Hz and (2) a maximum absolute slope in at least the 10th percentile. For Bird 5, the left EOG was used to calculate eye movement artifacts for all channels.

## Automated classification of sleep stages

We built on a previously published algorithm [28] that was originally designed to classify single-EEG sleep recordings in zebra finches and had been validated against manual REM/NREM scores. For each bird, one "best" EEG channel was chosen. The following variables were calculated for each 3-s window sliding by 1-s: log(delta), gamma/delta, gradient of delta, gradient of gamma/delta, nPeaks, maximum absolute amplitude, and the standard deviation of the waveform. The gradient was calculated using the numpy.gradient function in the Python NumPy library. All variables were z-scored.

Two independent k-means clustering steps were then run using the scikit-learn Python library [124]. The first step separated SWS and non-SWS and was performed on the 6 variables log(delta), gradient of delta, gradient of gamma/delta, SD, nPeaks, and maximum amplitude. The cluster with higher log(delta) was designed SWS. The second step separated REM from NREM and was performed on the 5 variables gamma/delta, gradient of delta, gradient of gamma/delta, SD, and nPeaks. The cluster with higher gamma/delta was designated REM.

The 2 sets of k-means classifications were then used to assign 4 possible scores. Epochs classified as REM and non-SWS were scored as REM, while epochs classified and NREM and SWS were scored as SWS. If an epoch was classified as NREM and non-SWS, it was scored as IS. Finally, if an epoch received classifications of both REM and SWS, it was scored as an artifact. Scores were then smoothed using a 5-epoch rolling mean, where SWS = 0, IS = 1, REM = 2, then rounded to the nearest integer. This step could "fill in" artifact epochs if there were any non-artifact epochs nearby, leaving only 1–8 epochs per bird scored as artifact.

To evaluate the concordance between manual scores (3-s resolution) and automated scores (1-s resolution), we examined every third value of the automated scores. Concordance was determined as (number of same scores) ÷ (number of total scores). The overall concordance, REM/NREM concordance, and SWS/NSWS concordance were each calculated separately.

It should be noted that concordance values are dependent on the relative proportions of a given stage. For example, if the algorithm incorrectly scored 100% of epochs as IS in a bird with only 10% REM, the REM/NREM concordance would be 90%. Therefore, we also calculated Cohen's kappa, which controls for chance in agreement between 2 sets of scores [61,62]. Kappa ranges from 0 to 1 and would be 0 in the above example of IS-only scoring. We computed kappa using the scikit-learn Python library [124] and evaluated kappa values using standard benchmarks [62].

We also analyzed the performance of the automated classifier using a signal detection framework. For example, when analyzing the classifier's ability to detect REM, the true positives (TPs) or "hits" are calculated as the number of REM epochs the classifier correctly scored as REM. In contrast, false negatives or "misses" are REM epochs that the classifier incorrectly scored as NREM. This can be used to calculate the sensitivity:

$$sensitivity = \frac{TP}{TP + FN}$$

To calculate the specificity, we then calculate the number of true negatives (TNs; NREM epochs correctly scored as NREM) and false positives (FPs) or "false alarms" (NREM epochs incorrectly scored as REM):

$$specificity = \frac{TN}{TN + FP}$$

Perfect classification would have both a sensitivity and specificity of 100%.

For detecting REM, the automated algorithm had a sensitivity of 61.3% ± 4.6% and a specificity of 84.6% ± 3.5%. For detecting SWS, the algorithm had a sensitivity of 51.8% ± 13.2% and a specificity of 76.8% ± 5.8%. This suggests that for both of these stages, the classifier is more specific than it is sensitive—it tends to detect only a subset of REM/SWS, but the epochs it detects tend to be accurately identified.

## Constant light: Motion detection

Motion was detected in grayscale video frames by determining the number of pixels with frame-to-frame change over a threshold, which we set to 10% of the maximum possible grayscale value for a pixel. This threshold was effective in capturing the animal's movement while filtering out slight fluctuations in lighting and other minor sources of frame-to-frame change.

For each of the 3 birds that underwent the constant light experiment, motion values were extracted from continuous video, starting at the time of baseline sleep recording and spanning all 8–12 d of constant light and the subsequent return to normal LD conditions. Data were averaged across 10-min bins for visualization and periodogram analysis.

To examine the periodicity of the animals' movement patterns, we computed the Lomb-Scargle periodogram [125,126] in Python using the AstroPy library [127,128]. For each bird, periodograms were calculated over the constant light period and over one continuous LD period, either baseline or recovery depending on which lasted longer for a given bird. For Bird 2, the baseline LD period was used (4 d). For Birds 3 and 4, the recovery LD period was used (7 and 4 d, respectively). Periodograms were calculated over the range of periods from 10 to 30 h. A large peak at or near 24 h indicates an intact circadian rhythm.

## Experimental design and statistical analyses

Statistical analyses were carried out in Python. We analyzed changes in sleep composition over the 2 halves of the night with two-tailed paired $t$ tests on IS, SWS, and REM. To look at hour-by-hour changes in sleep structure, we calculated Pearson's correlation coefficient between hour and the sleep measure of interest. Differences between the 3 sleep stages (e.g., episode durations, average ultradian phase) were compared with a one-way ANOVA, followed by paired $t$ tests. To assess the effect of LL on nPeaks, we performed a two-way repeated measures ANOVA; the within-subject factors were vigilance state (sleep versus wake) and lighting condition (LL versus LD). Significance was set at 0.05. Throughout this paper, we report variance as mean ± SD unless otherwise noted.

To address the problem of multiple comparisons when assessing spectral differences between IS, SWS, and REM, we used Rüger's areas [129,130], following the method described in [122]. For each pair of sleep stages, a set of paired $t$ tests were computed over the whole frequency range from 0 to 55 Hz, for each bin of 0.33 Hz. Then, starting from the lower frequencies, all ranges of neighboring, consecutive frequency bins with $p < 0.05$ were identified. To qualify as a significant Rüger's area, a range must meet the following criteria: (1) 100% of bins are significant at the $\alpha = 0.05$ significance level; (2) at least 50% of bins are significant at the $\alpha = 0.025$ level (i.e., half of the standard significance threshold); and (3) at least 33% of bins are significant at the $\alpha = 0.0167$ level (i.e., one-third of the standard significance threshold). If these criteria were met, the entire frequency range was considered significant. For completeness we also noted trends, i.e., Rüger's areas that met the first criterion but not all three [131].

## Code accessibility

PSG scoring was performed in the custom-written GUI arfview, available at https://github.com/margoliashlab/arfview. An updated PSG scoring GUI compatible with binary files is

available at https://github.com/margoliashlab/bark. The Python library Resin, which was used for multitaper spectral analyses, is available at https://github.com/margoliashlab/resin. Custom code used for event detection, automated scoring, and quantification of circadian and ultradian rhythms will be made available on https://github.com/margoliashlab.

## Supporting information

**S1 Video. Long-duration REM with eye movements and twitches.** The video shows 100 s from the baseline night of Bird 1 in real-time speed. The bird moves through wake, drowsy, and the 3 major sleep stages, including 2 episodes of REM each >20 s in duration. One eye is visible throughout. NREM is accompanied by stillness and regular deep breathing. During REM, eye movements and twitches appear; major events are labelled. Also note the relative stillness leading into and out of the wake episodes. IS, intermediate sleep; NREM, non-REM; REM, rapid eye movement sleep; SWS, slow wave sleep.
(M4V)

**S1 Fig. Hypnograms of sleep in all birds.** Rows correspond to stage; each tick mark represents one 3-s epoch. Vertical gray lines indicate lights on and off. White gaps starting at approximately 2 h for Bird 1 recording and approximately 30 min for Bird 3 recording indicate missing data due to bird moving out of frame (Bird 1) or technical problems with video recording (Bird 3). Each individual showed somewhat unique sleep patterns, with Birds 1 and 4 exhibiting more and slightly deeper daytime sleep than other birds. In general, sleep was much more consolidated at night; daytime sleep was fragmented and most prevalent in the afternoon. The patterns of REM increase and SWS decrease across the night are also visible here. Raw scores are provided in S2 Data. IS, intermediate sleep; REM, rapid eye movement sleep; SWS, slow wave sleep; Unihem, unihemispheric sleep.
(PNG)

**S2 Fig. Average normalized power spectra of wake, drowsy, and unihemispheric sleep.** (a) Spectra of Wake and Drowsy compared to REM and SWS, plotted relative to IS. The pattern of REM is markedly similar to Wake and Drowsy. There were no statistically significant differences between Wake and Drowsy. The qualitatively slightly lower gamma and slightly higher delta during drowsiness are consistent with the description of drowsiness as an awake state with some (NREM) sleep-like characteristics. (b) Unihemispheric sleep, plotted relative to IS, with Drowsy shown for comparison. EEG from the awake hemisphere (corresponding to the open eye) is contrasted with the sleeping hemisphere (closed eye). Due to the small amount of US detected in this study, sufficient artifact-free US data was collected from 5 channels and 4 channels, respectively, from 3 birds. The awake hemisphere is most similar to Drowsy. (c) Unihemispheric sleep (sleeping hemisphere), plotted relative to Wake, with REM, IS, and SWS shown for comparison. The sleeping hemisphere is most similar to IS, except for very low power in the gamma band most similar to SWS. Data are provided in S1 Data (under Fig 4A–4C). EEG, electroencephalogram; IS, intermediate sleep; NREM, non-REM; REM, rapid eye movement sleep; SWS, slow wave sleep; US, unihemispheric sleep.
(PNG)

**S3 Fig. Associations between REM and period of ultradian rhythms.** The color of each dot corresponds to bird identity, with Bird 1 in the darkest shade and Bird 5 in the lightest. Dashed line, line of best fit as determined by least-squares linear regression. (a) Fast rhythm (average period = 60 s); a positive correlation with REM was observed. (b) Slow rhythm (average period = 29 min). Data are provided in S1 Data. REM, rapid eye movement sleep; TST, total

sleep time.
(PNG)

**S4 Fig. Principal component analysis of automated sleep scores.** PCA was performed on the five-dimensional space of log(delta), gamma/delta, gradient of delta, gradient of gamma/delta, and standard deviation, as described in Fig 8. Each column shows a different combination of the first 3 principal components. Note that for Birds 1 and 4, the first PC is plotted from positive to negative to facilitate comparisons between all 5 birds. The first 2 columns show IS clustering in the center of PC1, with SWS and REM fanning out to either end. The rightmost column shows SWS and REM falling along 2 orthogonal planes, with IS forming the transitional area in between. This pattern was least robust in Bird 2, which had the lowest percentage SWS. Purple denotes REM, blue denotes SWS, and green denotes IS. Data are provided in S1 Data. IS, intermediate sleep; PC, principal component; PCA, principal component analysis; REM, rapid eye movement sleep; SWS, slow wave sleep.
(PNG)

**S5 Fig. Sleep/wake scores at baseline and in constant light.** LD, baseline 13:11 light:dark cycle; LL, constant bright light. Rows correspond to state as determined by video scoring; each tick mark represents one 3-s epoch. Vertical gray lines indicate time of lights on and off in baseline LD; gray shading indicates dark period. Differences in unihemispheric sleep in Birds 2 and 3 were likely due to altered camera positioning in LL (see Methods). In LL, sleep was reduced, fragmented, and spread out across the day. Raw scores are provided in S2 Data. LD, light/dark; Left U, left-brain unihemispheric sleep; LL, constant light; Right U, right-brain unihemispheric sleep.
(PNG)

**S1 Data. Quantitative data for each figure.**
(XLSX)

**S2 Data. Raw video and PSG scores.** PSG, polysomnography.
(XLSX)

# Acknowledgments

We thank Daniel D. Baleckaitis for assistance with surgical techniques and developing headgear. Kyler J. Brown wrote code used in manual scoring and spectral analysis. Timothy P. Brawn provided invaluable advice on avian EEG surgical, recording, and scoring techniques; gave helpful feedback on early drafts of this manuscript; and developed the nPeaks measure for use in avian sleep analysis.

# Author Contributions

**Conceptualization:** Sofija V. Canavan, Daniel Margoliash.

**Formal analysis:** Sofija V. Canavan.

**Funding acquisition:** Sofija V. Canavan, Daniel Margoliash.

**Methodology:** Sofija V. Canavan.

**Project administration:** Daniel Margoliash.

**Resources:** Daniel Margoliash.

**Supervision:** Daniel Margoliash.

**Visualization:** Sofija V. Canavan.

**Writing – original draft:** Sofija V. Canavan, Daniel Margoliash.

**Writing – review & editing:** Sofija V. Canavan, Daniel Margoliash.

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
