## [Editor Report · Decision Letter 0]

17 Aug 2020

Dear Sofija, 

Thank you for submitting your manuscript entitled "Budgerigars, like songbirds, have complex sleep structure similar to that of mammals." for consideration as a Research Article by PLOS Biology.

Your revisions have now been evaluated by the PLOS Biology editorial staff, and I'm writing to let you know that we would like to send your submission out for re-review.

Please re-submit your manuscript within two working days, i.e. by Aug 19 2020 11:59PM.

Kind regards,

Roli

Senior Editor

PLOS Biology

---

## [Decision Letter · Decision Letter 1]

16 Sep 2020

Dear Sofija,

Thank you for submitting your revised Research Article entitled "Budgerigars, like songbirds, have complex sleep structure similar to that of mammals." for publication in PLOS Biology. I have now obtained advice from one of the original reviewers and have discussed their comments with the Academic Editor. 

Based on the reviews, we will probably accept this manuscript for publication, assuming that you will modify the manuscript to address the remaining points raised by the reviewers.

IMPORTANT:

a) Please attend to the remaining comments from reviewer #3.

b) Many thanks for the very thorough provision of underlying data in S1_Data. However, please could you include in it the data for Fig S3, and then cite this file in the relevant Figure legends (i.e. legends for Figs 2, 3, 4, 6, 7, 8, 9, S3)?

c) Please could you simplify (and increase the appeal of) your title? We suggest "Budgerigars have complex sleep structure similar to that of mammals." You would, however, need to explain clearly in the abstract that songbirds share these features.

d) In your Ethics Statement, please provide the relevant IACUC protocol approval number.

We expect to receive your revised manuscript within two weeks. Your revisions should address the specific points made by each reviewer. In addition to the remaining revisions and before we will be able to formally accept your manuscript and consider it "in press", we also need to ensure that your article conforms to our guidelines. A member of our team will be in touch shortly with a set of requests. As we can't proceed until these requirements are met, your swift response will help prevent delays to publication.

- a cover letter that should detail your responses to any editorial requests, if applicable

*Copyediting*

*Published Peer Review History*

*Early Version*

Sincerely,

Roli

Senior Editor,

rroberts@plos.org,

PLOS Biology

REVIEWER'S COMMENTS:

Reviewer #3:

[identifies himself as Franz Weber]

Overall, the authors well addressed all of my concerns. Here some remaining final questions/ concerns about the new analyses: 

-Fig. 3

The transition matrix in Fig. 3h is extremely helpful in understanding the dynamics of the sleep pattern. 

I'm still a bit puzzled about what the edges in the chord diagrams represent. Fig. 3f contains a single edge that represents both IS -> REM and REM -> IS? 

Naively, I would have assumed that all the outgoing edges of one node (state) correspond to one row in the transition matrix (which sums up to 100%). That is, the outgoing edges show the likelihood to go to either of the states (P[current state -> next state])

Minor: Some of the rows in the matrix don't sum up to 100%, which is probably the result of rounding errors. 

-Fig. 9

(c) Why are there no error bars for SWA under LD conditions? What's the difference to the SWA data shown in Fig. 4d (which does have error bars, but seems to be slightly different)?

(c) What's the REM latency under LL?

What means 'pd' in 'LD mean sleep pd duration'?

-Fig. 6e:

What means the y-axis? I don't understand the "40% REM", "100% IS", and "20% SWS"? Would it be possible to plot for each state a separate line showing its rate or probability as function of the phase?

-Fig. 8

What about PC1 vs. PC3 (or PC2); do the data points for REM, IS, SWS lie along PC1 as in Low et al. 2008?

There are some "left over fonts" in (b, c) between "delta" and "-4". And quite a few points are located outside the 3D axes.

-Line 1276: "frequency bins with p > 0.05 were identified." Do you mean p < 0.05?

-Line 573: "many epochs identified as IS that did not collapse with either treatment?" What treatment?

-What is the rate of false positives and false negatives for REM, IS, SWS when comparing manual and automatic scoring? It may provide a more intuitive measure for how well manual and automatic scoring match.

---

## [Editor Report · Decision Letter 2]

8 Oct 2020

Dear Dr Canavan,

On behalf of my colleagues and the Academic Editor, Gilles Laurent, I am pleased to inform you that we will be delighted to publish your Research Article in PLOS Biology. 

Early Version

PRESS 

Kind regards,

Alice Musson

Publishing Editor, 

PLOS Biology

on behalf of

Roland Roberts,

Senior Editor

PLOS Biology